# China's electric vehicle and climate ambitions jeopardized by surging critical material prices

Hetong Wang[1,10], Kuishuang Feng [2,10], Peng Wang[3,4,5] ✉, Yuyao Yang[6], Laixiang Sun [2,7] ✉, Fan Yang[8], Wei-Qiang Chen [3,5], Yiyi Zhang [9] & Jiashuo Li[1] ✉

The adoption of electric vehicles (EVs) on a large scale is crucial for meeting the desired climate commitments, where affordability plays a vital role. However, the expected surge in prices of lithium, cobalt, nickel, and manganese, four critical materials in EV batteries, could hinder EV uptake. To explore these impacts in the context of China, the world's largest EV market, we expand and enrich an integrated assessment model. We find that under a high material cost surge scenario, EVs would account for 35% (2030) and 51% (2060) of the total number of vehicles in China, significantly lower than 49% (2030) and 67% (2060) share in the base-line, leading to a 28% increase in cumulative carbon emissions (2020-2060) from road transportation. While material recycling and technical battery innovation are effective long-term countermeasures, securing the supply chains of critical materials through international cooperation is highly recommended, given geopolitical and environmental fragilities.

Affordable electric vehicles (EVs) are seen as pivotal tools for achieving sustainable transportation by the mid-21st century[1]. However, a recent surge in the prices of critical materials (e.g., lithium, cobalt, nickel, and manganese)[2–5] in power batteries has led to widespread concerns on the competitiveness of EVs in the near future. For example, nickel price has been very volatile in 2022. On 8 March 2022, it topped $100,000 per ton before the London Metal Exchange (LME) was forced to step in and halt trading for the next few days, which "has never happened before in the history of the nickel market[6]". Some hedge funds argued that the LME's decision constituted an injury to their own rights and interests, and they wanted to seek compensation. However, LME emphasized that its decision had taken due regulatory process into account and was in the interest of the market as a whole. Although the nickel price has retreated from this peak, it is still relatively high. This type of volatility not only makes the market trend difficult to predict, but also puts great pressure on the EV market, which depends on lithium-ion battery (LIB). According to Bloomberg's report, the price of these critical minerals grew by 280% in 2021[7]. In the first quarter of 2022, prices for lithium alone have grown 438%[8]. Elon Musk, Tesla's chief executive, emphasized that prices of critical materials (lithium, cobalt, nickel) have reached insane levels, and Tesla may have to get into mining and refining directly on a large scale unless prices reduce[9,10]. In this context, there is an urgent need to assess the extent to which the material price surge could distract the uptakes of EVs, therefore helping formulate cost-effective EV deployment strategies.

[1]Institute of Blue and Green Development, Shandong University, 264209 Weihai, China. [2]Department of Geographical Sciences, University of Maryland, College Park, MD 20742, USA. [3]Key Lab of Urban Environment and Health, Institute of Urban Environment, Chinese Academy of Sciences, 361021 Xiamen, China. [4]Ganjiang Innovation Academy, Chinese Academy of Sciences, 341000 Ganzhou, China. [5]University of Chinese Academy of Sciences, 100864 Beijing, China. [6]Guanghua School of Management, Peking University, 100871 Beijing, China. [7]School of Finance & Management, SOAS University of London, London WC1H 0XG, UK. [8]Department of Planning, Aalborg University, 9000 Aalborg, Denmark. [9]Guangxi Key Laboratory of Intelligent Control and Maintenance of Power Equipment, Guangxi University, 530004 Nanning, China. [10]These authors contributed equally: Hetong Wang, Kuishuang Feng. ✉e-mail: pwang@iue.ac.cn; LSun123@umd.edu; lijiashuo@sdu.edu.cn

The literature on long-term supply and demand analysis of these critical materials suggests persistent and deepening shortage of their supply. According to the International Energy Agency (IEA)[11], at least 30 times as much lithium, nickel, and other key minerals would be required by the EV industry by 2040 to meet global climate targets, which far outstrips the committed mine production of these minerals. This stream of literature has shown that mineral shortage will constrain the deployment of EVs in the coming decades[12–14]. However, these studies have followed a material-flow perspective and treated the volumes of EV uptake as being largely independent of cost comparison between EV and its alternatives. In this research, we endogenize the uptake of EV as a result of cost comparison across all available options.

As the world's largest emitter of carbon dioxide ($CO_2$)[15], China has set an ambitious carbon neutrality target[16–18], which requires an 80% reduction of $CO_2$ emissions from transportation by 2050 compared to the 2015 level[19]. EVs have the potential to mitigate $CO_2$ emissions by replacing fossil-fuel-powered internal combustion engine vehicles (ICEVs)[20], thus being widely regarded as an indispensable component for the carbon neutrality pledge. According to China's pledge, by 2035, EVs will become the mainstream of new vehicle sales and the passenger sector will be fully electrified[21]. This means that China will continue to be the world's largest EV and battery producer and consumers in the coming decades and therefore, assessing the impact of material price on the vehicle fleet electrification in China has significant implications for achieving the carbon neutral target of the world.

A large number of studies have evaluated the positive impacts of cost reduction in low-carbon technologies (e.g., solar photovoltaics, wind, carbon capture and storage, and battery storage) on decarbonization towards various climate change mitigation goals[22–24]. Following this literature, most of projections on future EV uptakes assume that the cost of low-carbon technologies will continue to decrease with the increase of production scale[25]. However, this assumption is prone to the challenge that the price of critical materials needed for low-carbon technologies may surge due to the imbalance of supply and demand in the future. For example, it is widely acknowledged that the cost of LIB technology, which has been extensively used in EVs, would continue to decline to below $100/kWh by around 2030[1,26–28]. While it is true that technical innovation and economies of scale will continue to drive down the manufacturing costs of LIB[29], the material cost of LIB will be mainly driven by the level of scarcity given the limited reserve and declining ore grades of lithium, cobalt, and nickel[30,31], which are critical materials in current battery technologies. The Bloomberg New Energy Finance reported that doubling lithium cost could increase the cost of nickel cobalt manganese ($LiNi_{0.333}Co_{0.333}Mn_{0.333}O_2$) NCM111 battery by 8%[32]. A negligence of such an important cost factor may lead to a biased estimation of China's EV development in the future, which in turn affects the delivery on carbon neutrality commitment[33–35]. This study intends to fill this important niche in the context of China.

In order to effectively assess the impacts of the price surge of critical materials on EV uptake and then on road-transportation $CO_2$ emissions under the background of China's carbon neutrality commitment by 2060, we extend the Global Change Assessment Model (GCAM v5.2) to incorporate changes in the prices of lithium, cobalt, nickel and other critical materials. Previous studies do not consider the critical role of metals as inputs and the potential rise in costs due to the energy transition. This extended GCAM model allows us to quantitatively and consistently assess the dynamics of critical material price, the competitiveness of EVs with reference to alternative technologies, and the corresponding carbon emissions. It allows us to investigate EV development with different types of LIBs (nickel cobalt manganese ($LiNi_xCo_yMn_zO_2$) (NCM111, NCM622, NCM811, and NCM9.5.5), nickel cobalt aluminum ($LiCo_{0.15}Al_{0.05}O_2$) (NCA), lithium iron phosphate ($LiFePO_4$) (LFP), and lithium manganese oxide ($LiMn_2O_4$) (LMO)) and under different price surge scenarios of critical materials (lithium,

cobalt, nickel, and manganese). Our findings demonstrate that the price surge of critical materials will jeopardize the fleet electrification and put additional pressure on China's carbon-neutral ambition. Material recycling and technical innovation of LIBs are promising solutions in addressing the material price challenge, especially in the long term.

## Results

### Impacts of critical materials price surge on the future costs of electric vehicle (EV)

Figure 1a, b and Fig. S3 in Supplementary Information present the changes in the prices of critical materials (i.e., lithium, cobalt, nickel, and manganese) with reference to the 2015 levels under the High, Medium, and Low scenarios of material price surge over 2020–2060. Our results indicate that under the High scenario, cobalt would have the highest price increments of 213% by 2030, and 467% by 2060. The price of lithium would continue to rise over the first 15 years, rising by about 380% by 2035 and then remain stable. Nickel would experience a lowest price increment of 164% by 2060. Manganese, a key material that has relatively low material intensity, would elevate its price by 313% by 2060. Under the Medium and Low scenarios, prices of lithium, cobalt, nickel, and manganese would increase by 230%, 257%, 142%, and 116% (Medium); and 170%, 201%, 83%, and 121% (Low) by 2060, respectively, which are lower than those in the High scenario. The large increases in the prices of these critical materials could have a significant impact on the costs of batteries and thus cost of EVs.

Figure 1c and Fig. S6 in Supplementary Information show the EV cost evolution as driven by material price surge. In the absence of material price surge (the base-line scenario, BLS), future EV costs would continue to decline, mainly as a result of technical innovation. In sharp contrast, the introduction of material price surging will lead to a substantial increase in EV cost. Taking EVs equipped with nickel cobalt manganese ($LiNi_{0.6}Co_{0.2}Mn_{0.2}O_2$) (NCM622) LIBs as an example, under the High scenario, the light duty vehicle-four wheels (LDV-4W) sector would have the highest increment in EV cost, which would reach 0.046, 0.070, 0.106, and 0.141 (1990)$/pass-km for a mini car, a subcompact car, a compact car, and a large car and SUV by 2030 (7%, 9%, 10%, and 8% higher than those in the BLS scenario, respectively), with the corresponding cost figures reaching 0.048, 0.073, 0.116, and 0.148 (1990) $/pass-km by 2060 (15%, 19%, 21%, and 18% higher than those in the BLS scenario, respectively). The EV cost in the bus sector will also increase sharply, making the cost of the light bus and heavy bus reach 0.023 and 0.040 (1990)$/pass-km by 2060 (11% and 15% higher than those in the BLS). The electric trucks sector would have the lowest increment of about 9% compared to the BLS scenario by 2060. Meanwhile, the EV cost under the Medium scenario would be about 3–12% lower than that in High scenarios by 2060, due to the relatively lower level of threat by material price surge. The extent of EV cost increases will also be further reduced under the Low scenario, but from 2035 onwards, it would be 1–6% higher than those in the BLS. This set of result clearly shows that if the prices of critical materials continue to surge, the cost of EVs would be elevated by significant margins.

The costs of those EVs equipped with other types of LIBs will also be driven up by the price surges of critical materials, just like EVs with NCM622 LIBs (Figs. S5–S11). Under the High scenario, their costs would continue to rise, especially after 2035, with a relatively high increase for EVs equipped with ternary LIBs (5–14% and 5–32% higher than those in the BLS scenario by 2030 and 2060, respectively). The costs of those EVs equipped with cobalt-free LIBs (LFP and LMO) would have a smaller increase by about 3% (by 2030) and 4% (by 2060) compared with the BLS scenario. While the increase in EV cost is obvious across all critical material price surge scenarios, especially in the long term, the magnitude is smaller in Medium scenarios (11–17% lower than that in High scenario by 2060) and Low scenarios (12–18% lower than that in High scenario by 2060). On the other hand, it is worth noting that the

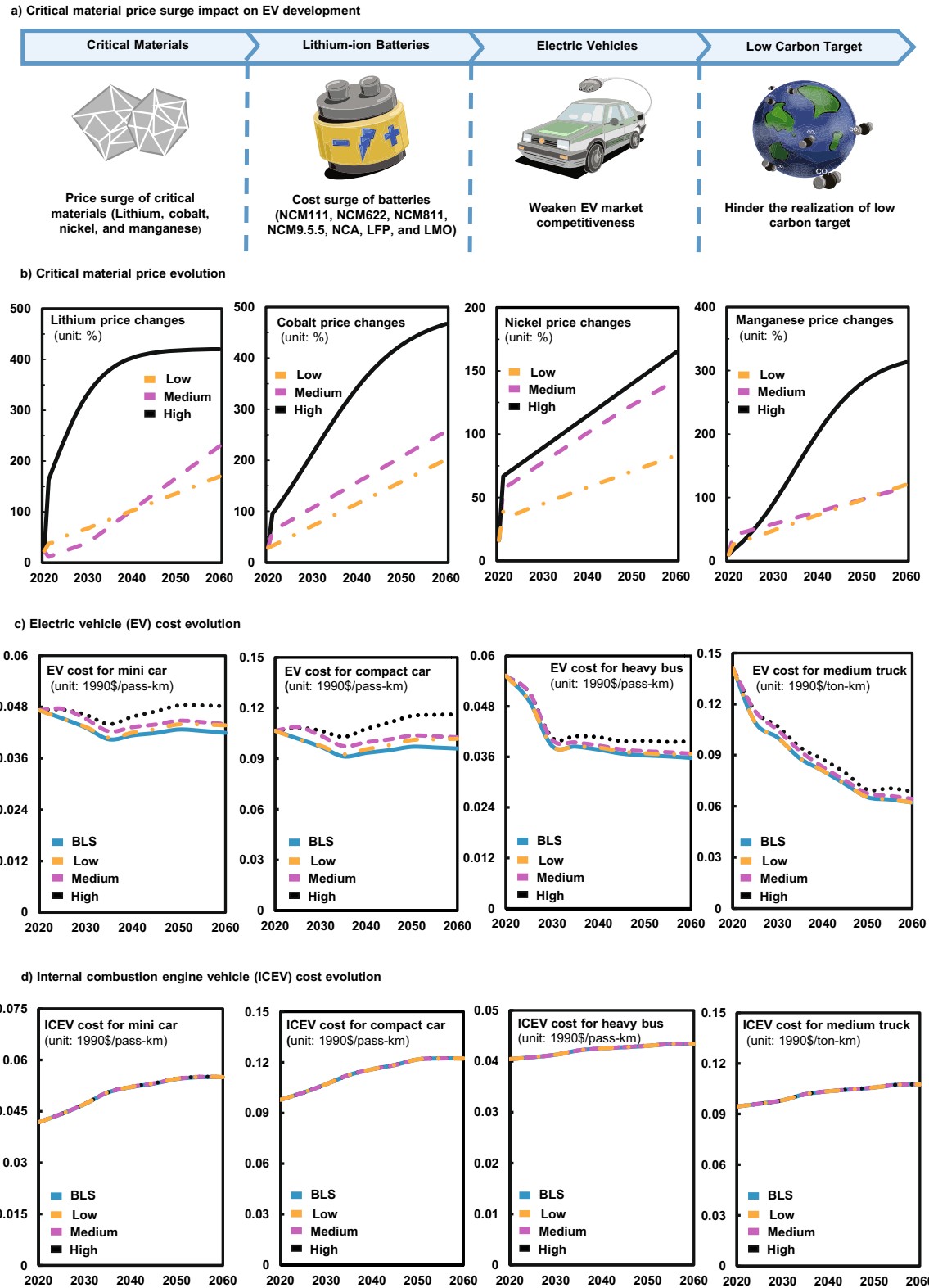

**Fig. 1 | Evolution of critical material prices and costs of EVs and ICEVs from 2020 to 2060 under different scenarios. a** The illustration of the key cause-effect links of the study, **b** Price evolution of the four critical materials over 2020–2060, **c** Cost evolution of EV by sub-sector, **d** Cost evolution of ICEV by sub-sector. BLS refers to the base-line scenario in which the uptake pace of EVs will fulfil the requirement of the carbon neutrality target and the EV cost will fall rapidly in line with its historical and forecasted development trend in China as reported in the existing literature; High scenario in which a rapid increase in critical material price affects EV costs; Medium scenario in which a steady increase in critical material price affects EV costs; Low scenario in which a slight increase in critical material price mainly affects EV costs during the middle and later periods of the forecast. All EVs in this figure are equipped with NCM622 LIBs; and the change in prices of critical materials is compared to the corresponding prices in 2015.

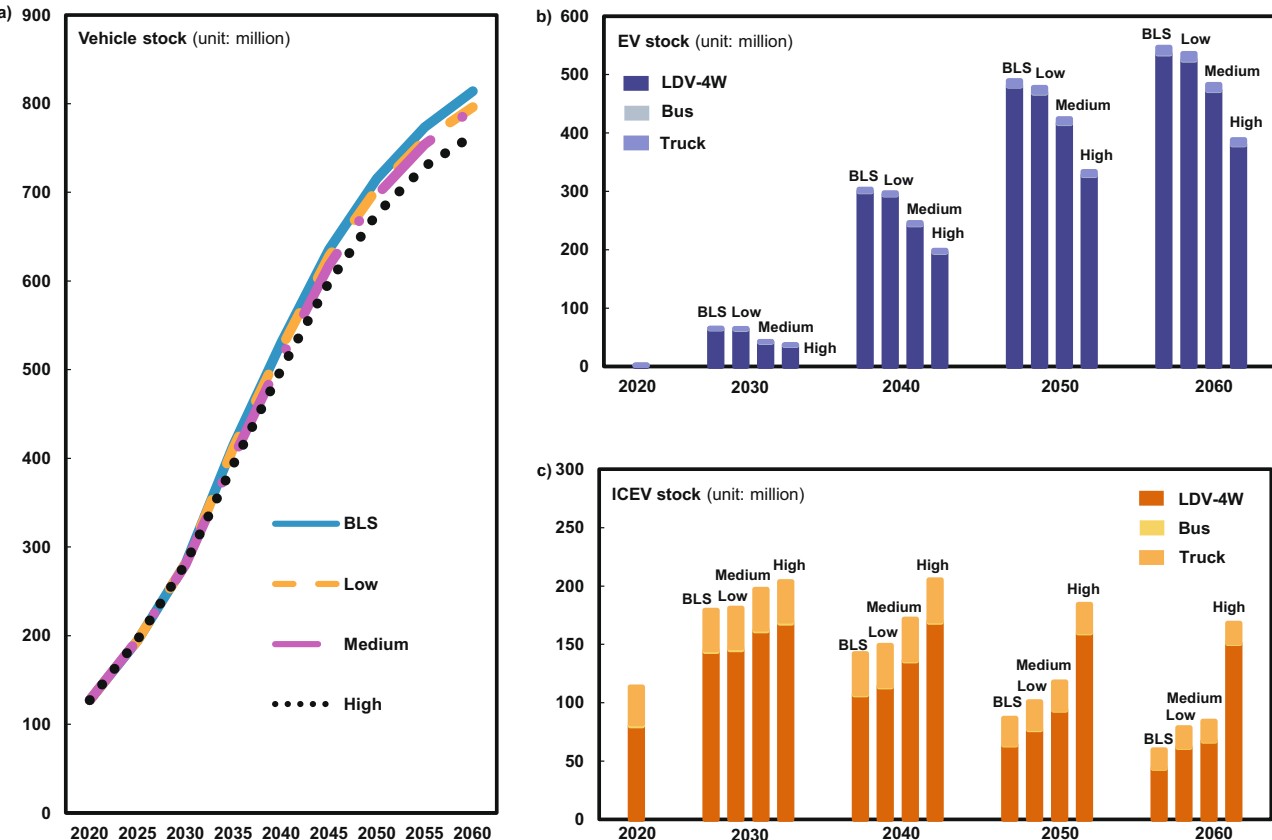

**Fig. 2 | Projections of vehicle stocks through 2020 to 2060 under different scenarios. a** Total vehicle stocks, **b** EV stocks by subsector, **c** ICEV stocks by subsector. The scenarios of BLS, High, Medium, and Low are the same as in Fig. 1; EV, electric vehicle (electric vehicle refers to battery electric vehicle in this paper); ICEV, internal combustion engine vehicle; All EVs in this figure are equipped with NCM622 LIBs.

costs of EVs equipped with high-cobalt LIBs would be greatly affected by the surging price of critical materials (lithium, cobalt, nickel, and manganese) in LIBs.

It is the relative costs that influence the choice of consumers between competing technologies (e.g., ICEV). Therefore, we also analyze the evolution of ICEV cost (Fig. 1d and S12). Under BLS scenario, the ICEV costs in the LDV-4W sector are 0.042, 0.058, 0.098, and 0.153 (1990)$/pass-km for a mini car, a subcompact car, a compact car, and a large car and SUV in 2020, and the increments of these values will be 4–12% by 2030 and 6–31% by 2060. The ICEV costs for light bus and heavy bus will be 0.027 and 0.043 (1990)$/pass-km, respectively, by 2060, which are about 5% higher than those in 2030. The ICEV costs in the truck sector will increase by 1–11% between 2030 and 2060. The ICEV costs under the material price surge scenarios show slightly increases by about 0.02–0.05% from the BLS, as a result of the increase in fuel costs caused by consumers switching from EVs to ICEVs. These results suggest that the ICEV cost will remain relatively stable during 2020 to 2060 under all scenarios.

**Price surge of critical materials weakens EVs penetration rate**
Figure 2a shows that the projected total stocks of China's EVs equipped with NCM622 LIBs would reach about 66 million units (25% of the total vehicle stock) by 2030, and around 550 million (68% of the vehicle stock) by 2060 under the BLS carbon neutral scenario. Figures S19-21 present the results for EVs equipped by all types of batteries under all four scenarios. The results under the BLS indicate that the cost reduction through technical innovation and in the absence of material price surge would accelerate fleet electrification. However, the increased cost of EVs caused by critical material prices surging would undermine the positive effect of technical innovation and hinder the

progress of fleet electrification. As shown in Fig. 2b, there would be around 37 million (High), 43 million (Medium), and 65 million (Low) units of EV in China by 2030, which are 44%, 35%, and 1% lower than those in the BLS in 2030, and these shares will decrease to 29%, 12%, and 2% by 2060. The increase in the cost of EVs makes ICEVs more economically attractive. As a result, the total stock of ICEVs would reach 204 million (High), 197 million (Medium), and 181 million (Low) units by 2030, which is 6, 5, and 3 times the corresponding EV stock, respectively (Fig. 2c).

Figure 3 and Figs. S22–28 confirm that the predictions of future EV penetration agree with the increasing market share of EVs[36,37]. In the absence of material price surge (BLS scenario), the penetration rate of EVs will continue to increase, reaching a plateau of about 71% around 2045 and then starting a moderate decrease between 2055 and 2060 and end at 67% by 2060 (due to the increased adoption of hydrogen fuel vehicles (FCEVs)) (Figs. S22–28). The surges of critical material prices tend to decelerate this penetration trend. Taking EVs equipped with NCM622 LIBs as an example, the EV penetration rate would decline to 35%, 41%, and 43% by 2030 under the High, Medium, and Low scenarios, respectively. Due to the continuous surge in the prices of critical materials, the resulting penetration rates of EVs under the High, Medium, and Low scenarios would be reduced to 51%, 60%, and 66%, respectively, in 2060, being 24%, 11%, and 1% lower than those under the BLS. With the increase in EV cost, the penetration rate of ICEVs will increase by 14 and 16 percentage points under the High scenario compared with the BLS by 2030 and 2060, respectively (Fig. S23). In each of the LDV-4W, bus, and truck sector, the EV penetration rate in the LDV-4W sector is more forcefully influenced by the price surge of critical materials, largely because EVs are mainly used for passenger vehicles, while commercial vehicles prefer to adopt FCEVs.

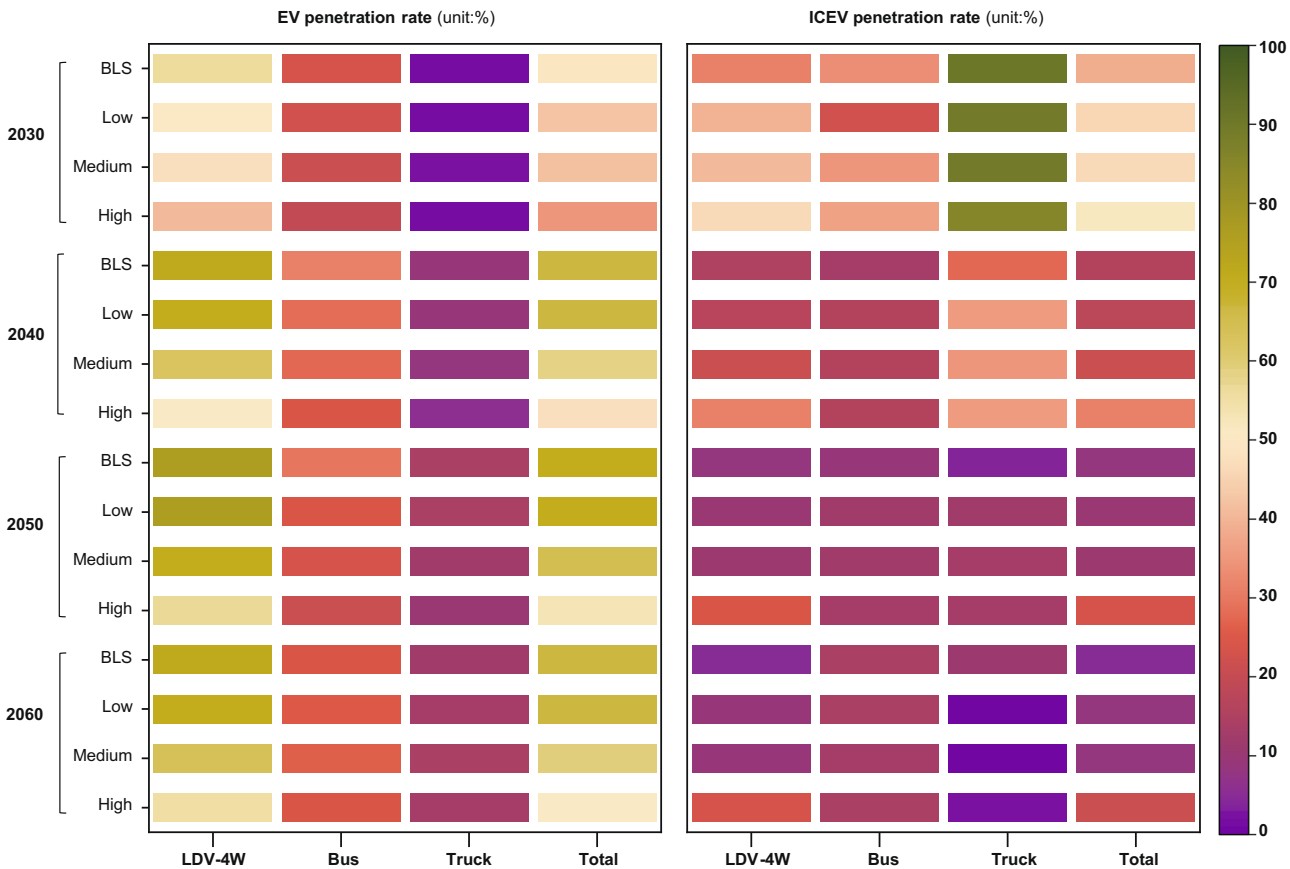

**Fig. 3 | EVs and ICEVs penetration rate.** EV, electric vehicle (electric vehicle refers to battery electric vehicle in this paper). ICEV, internal combustion engine vehicle. The scenarios of BLS, High, Medium, and Low are the same as in Fig. 1.

These results suggest that the surge in the prices of critical material would weaken the market competitiveness of EVs by significant margins.

Let's further explore the changes in EV penetration rates for those cases that EVs are equipped with different LIBs (Fig. 3 and Figs. S22–28). Our results show that the penetration rate of EVs would be 5, 12, and 13 percentage points higher under the NCM622-High, NCM811-High, and NCM9.5.5-High scenarios than that in the NCM111-High scenario by 2030, and the corresponding values would increase to 22, 33, and 37 percentage points by 2060. This means that replacing costly cobalt with nickel in LIBs can improve the market competitiveness of EVs. The cobalt-free LIBs will only increase the ICEV penetration rate by about 5 (2030) and 3 (2060) percentage points under the LFP-High and LMO-High scenarios, and the resultant ICEV penetration rate are 9 and 13 percentage points lower than those in the NCM622-High scenario, respectively. These results also suggest that developing batteries which do not contain extremely scarce materials would be able to mitigate the threat of critical material price surge to EV penetration rate by a significant extent.

**Cost surge increases carbon dioxide (CO₂) emissions**
Figure 4 and S30 report the direct (i.e., tailpipe) CO₂ emissions from road transport for 2020–2060 under different EV penetration conditions. Thanks to the extensive deployment of EVs under the BLS scenario, CO₂ emissions from road transport would reach a peak of 0.63 Gt/yr by 2030 and then declining to 0.22 Gt/yr by 2060. The main emission source in this case is the LDV-4W sector, which is responsible for over 60% of the road transport CO₂ emissions over the forty years. However, as soon as the surge in material price pulls down the EV penetration rate, the peak value of CO₂ emissions under

the High scenarios would become about 10% higher than that under the BLS scenario by 2030, reaching 0.66–0.71 Gt/yr. By 2060, CO₂ emissions would become 1.2 and 2.5 times that under the BLS scenario, being 0.27–0.54 Gt/yr. The reason is that the increasing penetration rate of ICEVs would elevate the demand for fossil fuels (Figs. S13–18), leading to an increase in CO₂ emissions. In this case, the passenger sector tends to use EVs and therefore would be more vulnerable to the surge in the prices of critical materials, resulting in the highest CO₂ emission increment (accounting for ~75% of total CO₂ emission in 2060) in comparison with other scenarios, with the LDV-4W sector, in particular, being the fastest growing sector for CO₂ emissions (Fig. 4b, d, f). These results indicate that decreasing market penetration rate of EVs relative to that under the BLS scenario would lead to higher CO₂ emissions in China's road transportation sector, especially from the LDV-4W sector.

Another set of findings indicates that the increment of CO₂ emissions is greater for EVs equipped with ternary LIBs, especially those with high-cobalt LIBs. First, with the reduction of cobalt content, the CO₂ emission under the NCM622-High, NCM811-High, and NCM9.5.5-High scenarios would be 22%, 35%, and 39% lower than those under the NCM111-High scenario by 2060. Second, the increment of CO₂ emission under the Medium and Low scenarios would be about half that under the High scenarios due to the smaller extent of material price surge pressure on EVs. The use of cobalt-free LIBs could relieve the pressure on CO₂ emissions, with CO₂ emissions of about 0.66 Gt/yr by 2030 and 0.27 Gt/yr by 2060 using LFP or LMO LIBs, being 3–7% by 2030 and 18–50% by 2060 lower than those using NCM LIBs under High scenarios. As a result, exploring alternative materials for making batteries would be able to reduce the cost pressure on EV development.

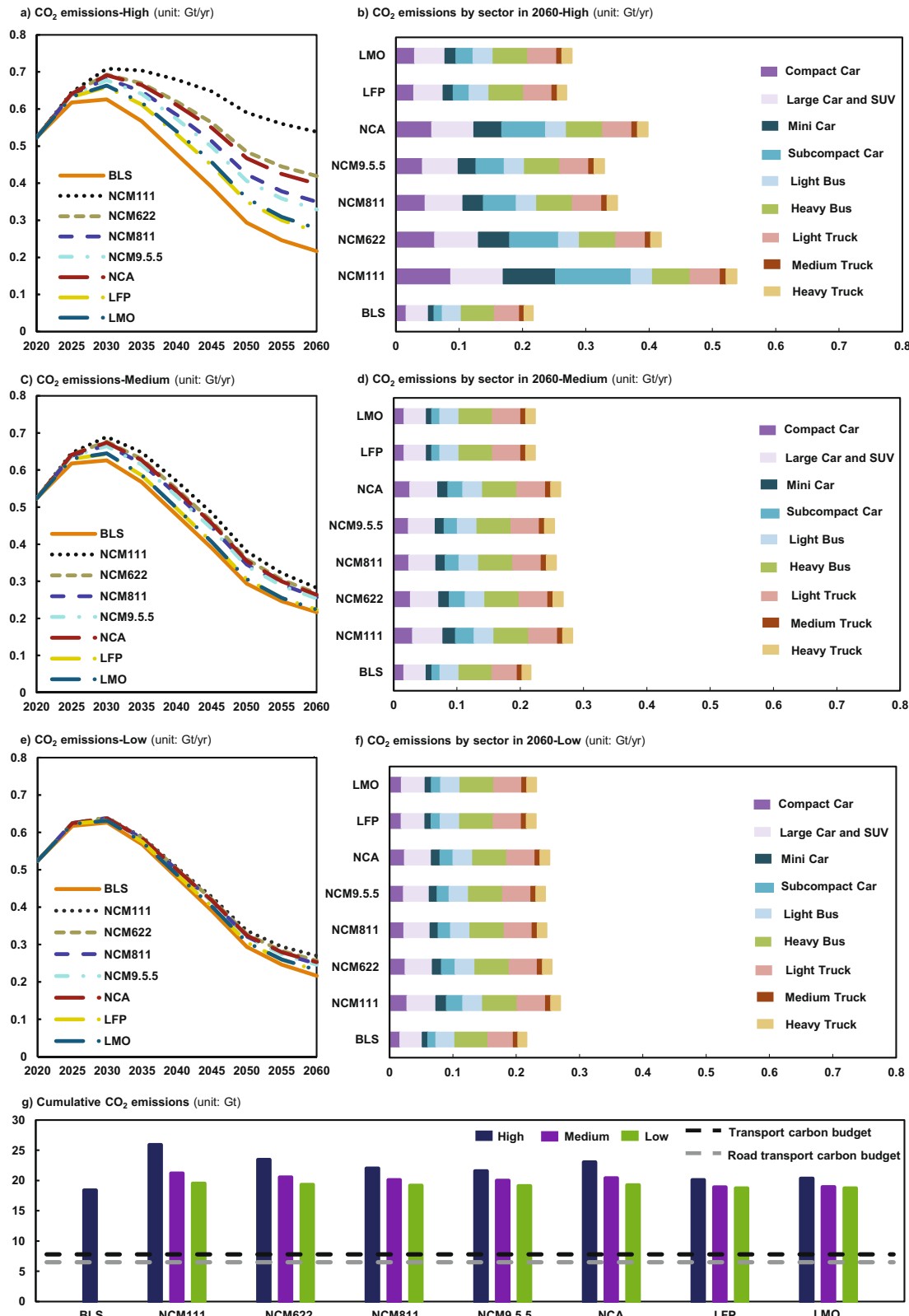

**Fig. 4 | CO₂ emissions of road transportation in China from 2020 to 2060.** **a**, **c**, **e** CO₂ emissions by 2060 under the High, Medium, and Low scenarios, respectively. **b**, **d**, **f** road transport's CO₂ emissions by sector by 2060 with EVs being equipped with different lithium-ion batteries (LIBs). **g** the cumulative CO₂ emissions through 2020–2060. The scenarios of BLS, High, Medium, and Low are the same as in Fig. 1.

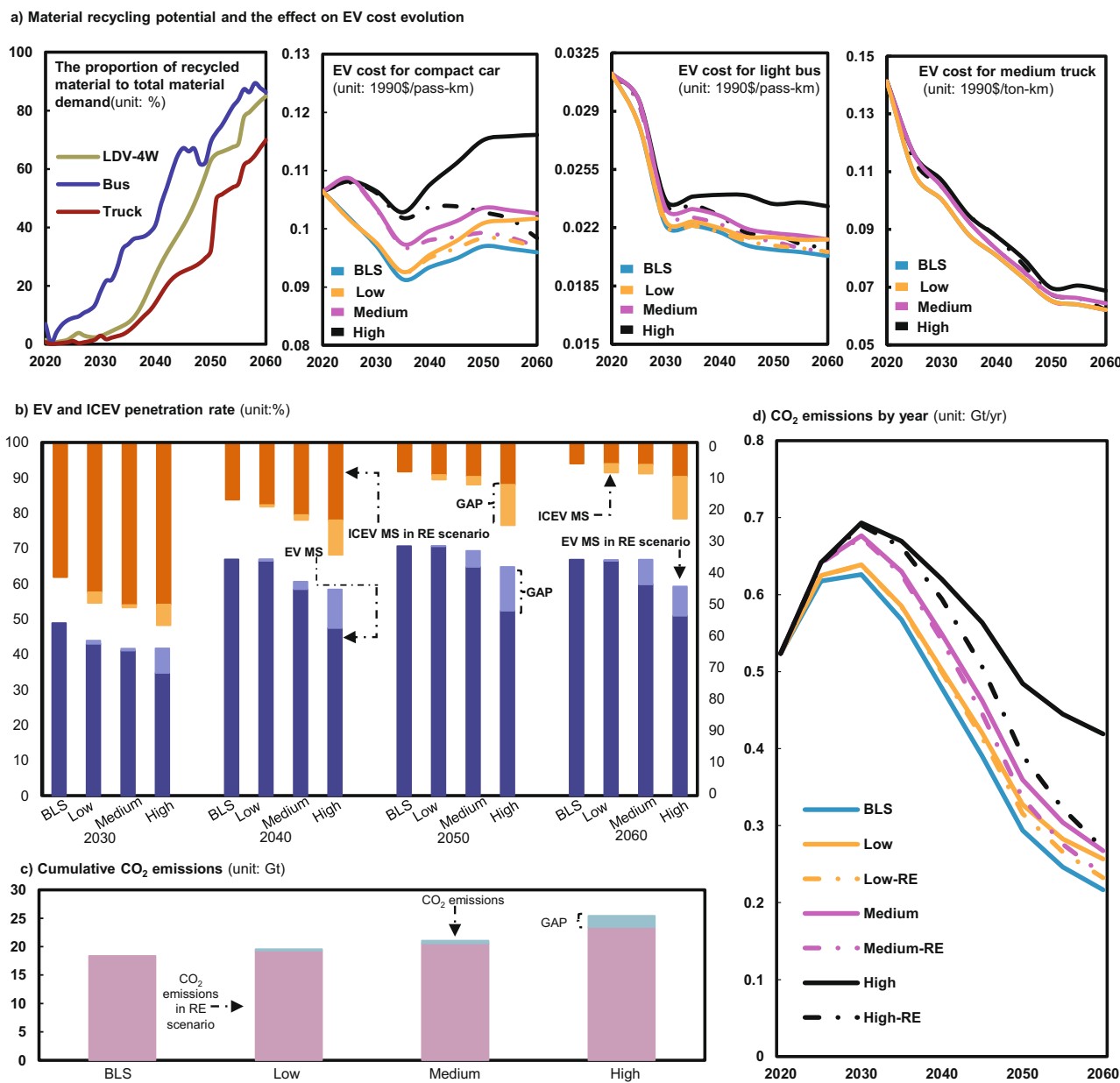

**Fig. 5 | Material recycling effects on EV development and CO₂ emissions of road transportation in China from 2020 to 2060 under the RE scenario. a** Material recycling potential and the effect on EV cost evolution. **b** EV and ICEV penetration rate by 2060. **c** Cumulative CO₂ emissions. **d** CO₂ emissions by year. *Note*: The scenarios of BLS, High, Medium, and Low are the same as in Fig. 1. RE is a scenario in which only primary demand is affected by the market price of the material concerned; MS is the market share of vehicles; EV, electric vehicle (electric vehicle refers to battery electric vehicle in this paper); ICEV, internal combustion engine vehicle; LDV-4W, light duty vehicle-four wheels; All EVs in this figure are equipped with NCM622 lithium-ion batteries (LIBs).

Figure 4g exhibits the cumulative CO₂ emissions from 2020 to 2060 in road transportation under different scenarios. Our result shows that when EVs are equipped with NCM622 LIBs, the cumulative CO₂ emissions in road transportation could reach 23, 21, and 19 Gt under the High, Medium, and Low scenarios, respectively, which will be 28%, 12%, and 5% higher than those under the BLS scenario. These values are 3.6, 3.2, and 3.0 times the carbon budget of the road transport sector, and 3.0, 2.6, and 2.5 times the transport carbon budget. EVs with cobalt-free LIBs (LFP and LMO) could reduce cumulative CO₂ emissions from road transport to a level of about 19 Gt, meaning a decrease by 3%-14% compared with the NCM622 scenario. The above discussion indicates that the material price surging in EVs would put new pressure on China's effort to reduce CO₂ emission in road transportation and even undermine the achievement of China's carbon neutrality goal by 2060.

## Material recycling promotes fleet electrification

Figure 5 report the results under the combinations of the RE (recycling) scenario and the High, Medium, and Low scenarios. The recycling potential of materials shows an increasing upward trend (Fig. 5a). Due to the delayed effects of material recycling, the resulting proportion of recycled materials to the total material demand in the LDV-4W, bus, and truck sector will be only 3%, 18%, and 3%, respectively, in 2030, however, this value could reach 85%, 86%, and 70%, respectively, by 2060. The recycled materials reduce the extent to which the materials needed for EVs are exposed to material price surges on international markets, thus reducing the likelihood of cost surging for EVs. Taking EVs equipped with NCM622 LIBs as an example (Fig. 5a and S32), the EV cost will decrease to about 0.05–0.15 (1990)$/pass-km by 2030 for LDV-4W under High-RE, Medium-RE, and Low-RE scenarios, which are slightly lower than

those in High, Medium, and Low scenarios, respectively. In the bus and truck sector, material recycling will only help decrease the EV cost by about 1% by 2030. But the benefits of material recycling can be significant in the long term. By 2060, the cost of EVs in the High-RE, Medium-RE, and Low-RE scenarios will be 11–15%, 4–6% and 4–5% lower than those in the High, Medium, and Low scenarios in the LDV-4W sector to reach about 0.05–0.13 (1990)$/pass-km, which is basically the same as that in the BLS scenario (even in the High scenario). The EV cost in bus and truck sector will have about 2–10% decrease in RE scenarios by 2060. Other EV cost evolution in RE-combining scenarios can be found in Figs. S31–S37. These results clearly manifest that material recycling can greatly reduce the impact of surging material prices on the EV cost, especially in the long term.

The decrease in EV cost will raise EV penetration rate. As shown in Fig. 5b, the material recycling will help increase the EV (with NCM622 LIBs) penetration rate by 7, 1, and 1 percentage points by 2030 in High-RE, Medium-RE, and Low-RE scenarios, respectively, and the resultant rates are still 14%, 8%, and 6% lower than those in BLS scenario. However, the recycling will boost the EV uptake rate to 59%, 66%, and 67% in High-RE, Medium-RE, and Low-RE scenarios, respectively, by 2060 (much closer to 67% under the BLS). This will inevitably reduce ICEV's market share by 12 percentage points (to about 10%), 2 percentage points (to about 6%), and 2 percentage points (to about 6%), making ICEV's market share close to that in the BLS (6%). Let all EVs be equipped with cobalt-free LIBs, recycling can reduce ICEV market penetration to baseline levels even in the High-RE scenario (other RE-combining scenarios than NCM622-RE can be found in Figs. S38-S44). These results highlight that recycling would have remarkable effects to mitigate the material price challenge in EV development in the long term.

Due to the positive role of material recycling in promoting EV development, the resulting cumulative $CO_2$ emissions from road transportation in 2020 to 2060 under High-RE, Medium-RE, and Low-RE scenarios decrease to 22 Gt, 20 Gt, and 19 Gt, respectively, which are 8%, 2%, and 1% lower than those under High, Medium, and Low scenarios, respectively (17%, 9%, and 4% higher than those in BLS scenario) (Fig. 5c). Although the $CO_2$ emissions from road transportation in the RE-combining scenarios will only decrease by less than 1% compared with that in High, Medium, and Low scenarios by 2030. After 2030, the differences in $CO_2$ emissions between the RE and BLS scenarios become narrowing. By 2060, the $CO_2$ emissions will decrease to 0.27 Gt/yr, 0.24 Gt/yr, and 0.23 Gt/yr in High-RE, Medium-RE, and Low-RE scenarios, respectively, which are 36%, 12%, and 10% lower than those in High, Medium, and Low scenarios (Fig. 5d). For EVs with cobalt-free LIBs, materials recycling can reduce the cumulative $CO_2$ emissions to a level only 2-7% higher than those under the BLS (Fig. S45). This indicates that materials recycling can facilitate low-carbon transition in the transportation sector in the long term.

## Discussion

In this paper, we have performed a detailed analysis of how the surge in the prices of critical materials could erode the adoption of EVs as required by the carbon neutrality target. We find that the previous estimates of EV development in China may be overly optimistic if the factor of critical material price surge is not considered[2,36–40]. The surge in the prices of critical materials could severely undermine the development of EVs, especially electric passenger vehicles, and make ICEVs more economically attractive on the market. This shift in favor of ICEVs would lead to a large increase in direct $CO_2$ emissions. For instance, the cumulative $CO_2$ emissions of the transportation under the NCM622-High scenario from 2020 to 2060 would be 28% higher than that under the BLS scenario. Such extent of increase in $CO_2$ emission may jeopardize the realization of China's carbon neutrality target by 2060.

The findings of this study highlight the high likelihood that both the EV development and carbon neutrality targets in China would be undermined by the increasing scarcity of various critical materials. In addition to the input requirement of EV development, critical materials are also needed for other low-carbon technologies. Examples include neodymium, dysprosium, and praseodymium in wind power generation[41]; germanium, tellurium, indium, gallium, and manganese in solar power generation[42,43]; nickel, cobalt, lithium, and platinum in fuel cell[2,44], and uranium, tungsten, tantalum, and molybdenum in nuclear energy[45]. This means that the EVs sector has to compete with other low-carbon technologies for critical materials. It is highly likely that this competition will push up the prices of these critical materials far beyond our current expectations. What makes the competition tougher is that a number of these materials are concentrated in a few countries in politically volatile regions and produced by a handful number of companies[46–48]. Geopolitical tensions and socioeconomic unrests in the producing regions would disturb the material supply and result in significant price volatility[49,50]. For example, cobalt is mined mainly as a by-product of nickel and copper, with approximately 71% of production and 51% of reserves concentrated in the Democratic Republic of Congo (DRC)[51]. In 2018, a policy shift in the country triggered an economic cascade that suspended the operations of Glencore's Mutanda mine, one of the DRC's largest cobalt mines. Whereafter the government announced to increase its mining royalty from 2% to 10%, price turbulence followed as a consequence[30]. The ongoing Ukraine–Russia crisis has also brought additional volatilities to the supply of critical materials[52]. How to ensure the supply security of critical materials is a great challenge to the EV sector in China and beyond.

Our analysis further shows that material recycling and technical innovation of battery chemistries can play very effective role in addressing the above challenge, especially in the long term. For example, the market share of EVs using cobalt-free LFP would be much larger than that of EVs using NCM batteries, and with material recycling, even in the High scenario, the penetration rate of EVs would not depart much from the baseline level. This result is consistent with the opinion of Sun et al.[53] i.e., the surging lithium price are not likely to impede the EV boom. But given the severity of the cobalt price challenge, we need to aggressively push materials recycling, especially in the long term, as recycling can reduce ICEV penetration rate by 6 percentage points by 2030 and 12 percentage points by 2060 under the NCM622-High scenario, making the ICEV penetration rate only 8% and 4% higher than those in the BLS.

Given the significant impact of material input price on EV adoption, market players across the supply chain and policy makers in China and other major player countries should make greater effort to manage the risk of material price surge. First and foremost, it is important to make the supply chains withstand the price surge caused by material supply[54]. While extraction technology continues to advance, mining costs are likely to rise as mineral quality declines and the carbon burden is monetized. Meanwhile, a substantial expansion of mining will inevitably produce adverse effects on the environment. That is, the supply of critical materials will be at high risk if volatility on commodity markets and environmental pressures on raw material extraction persist[55]. Manufacturers need to diversify production across the globe and maximize the extent of material recycling to build a long-term, sustainable and resilient supply chain[33,56,57]. Recycling is promising in addressing long-term critical material price challenges, as technological developments and economies of scale will reduce recycling costs. While recycling shortens supply chains and reduces logistical costs, at present it is still less expensive to mine the minerals than to recycle them, therefore, discovering processes for recovering valuable minerals which are cheaply enough to compete with newly mined minerals is urgently needed[58]. Open-loop secondary sources may be an ideal choice (e.g., manufacturing scrap) to meet the challenges of closed-loop material recycling (e.g., the technical constraints), as secondary sources are often more widely distributed across geographical space[59]. Nonetheless, the grade of recycled

material may require special attention. For example, nickel recovered from stainless steel is typically not in suitable quality for batteries due to the high iron content.

Second, technical innovation must be guided by forward-looking market information, with a special emphasis on developing alternative technologies which use less scarce materials (e.g., sodium-ion batteries[30]), on cathode improvements to reduce the content of precious materials (e.g., moving from NCM111 to NCM622, NCM811, and even NCM9.5.5)[60,61], and on exploring cobalt-free LFP technology[62]. However, these alternatives often face their own challenges, for instance, high-nickel NCM batteries heavily rely on carcinogenic nickel and LFP batteries require low energy storage efficiency iron[30]. Another concern is that it remains unclear whether the new technologies can meet the requirements of energy density, lifespan, cost-competitiveness, and safety of EV batteries. With regard to sodium-ion batteries (SIBs), the cost of SIBs is estimated to be about 10-20% less than that of their LIBs counterparts[63]. Chinese battery giant Contemporary Amperex Technology Co. Limited (CATL) are devoted to develop new SIB to help ease the potential cost pressure triggered by lithium price surge[64]. Although the promising cathodes do exist, the anode is the main bottleneck for the development SIBs[65]. Alternative types of electrodes based on cheap, common metals such as copper or iron fluoride and silicon, chemically bonded to store lithium ions, are perhaps the most promising candidates. But these alternatives must overcome problems related to stability, charging speed, and manufacturing[66]. In addition, with the spread of reuse and innovations in battery chemistry, battery life may be extended, which could reduce the potential for cost increases in battery chemistries, as this reduces the requirements for materials.

Third, addressing the material price surge risk of EVs is only possible in a favorable policy environment. Policy makers need to establish a cost-effective long-term mechanism to strengthen the control capacity and supply management of critical materials so as to counter their trade risk on the basis of understanding the demand, domestic reserves, recycling potential, and global production and trade pattern for critical materials. To better implement the medium- and long-term plan of mineral resource management, the government can take advantage of a range of tools at their disposal, including regulation, investment, subsidies, etc. Meanwhile, the new patterns of demand and material cost are likely to affect long-term macro-fiscal performance and policies, thereby raising strategic trade risks for critical materials of importing countries[39]. Governments and banks are also supposed to aim investment in recycling and reuse of critical materials to reduce dependence on imports.

Fourth, shared mobility schemes may help ease the growing desire for vehicle ownership and usage, thus indirectly reducing the demand for critical materials. Shared mobility schemes have the potential to reduce both personal vehicle usage and rates of ownership, which allows us to serve more users using less vehicles in a resource constrained world[67]. Recent research evidence shows that the experience of using car-sharing has a significant influence on decreasing the likelihood of choosing to use privately owned travel tools, such as private car[68,69]. Therefore, government agencies and private-sector transport operators need to work together to develop attractive pricing models, combined with awareness campaigns to encourage consumers to better participate in and understand shared mobility schemes. The sequent snowball effect would help cities reap the huge potential benefits of these new forms of mobility and help the EV sector to better cope with the constraint of material scarcity.

Last but not least, considering the relatively limited storage and production capacity[70] as well as the uneven geographical distribution[22] of critical materials, states will need to skillfully manage the dual relationships of cooperation and competition. In the long run, states may establish a global industry alliance in line with the United Nations'

Sustainable Development Goals to support the development of critical material industries and low-carbon technologies, thereby activating international supply chains, promoting a stable supply of critical materials, promoting resource recycling, and reducing system risks of material cost surges. In this regard, China, the EU, and the US, as the top players, need to take the lead.

## Methods

### Global Change Assessment Model (GCAM) framework

Global Change Assessment Model (GCAM)-v5.2 is a bottom-up, technology-rich integrated assessment model that depicts key inter-actions across economic, energy, land, water and climate systems. GCAM divides the world into 32 regions, with China being one of these 32 regions. GCAM-v5.2 can be openly accessed at https://github.com/JGCRI/gcam-core/releases and its documentation is available at http://jgcri.github.io/gcam-doc/v5.2/toc.html. GCAM-v5.2 has a flexible structure to develop heterogeneous sector structures in each region. Its Transportation Module includes the full spectrum of sub-modes and technologies available in passenger and freight transport and the corresponding input parameters which represent the real-world heterogeneity in a way consistent with the latest literature on transportation. GCAM-v5.2 models the endogenous interactions of transportation with other sectors within an individual region, as well as with other regions, and therefore, this integrated framework is well-suited for analyzing China's transportation development when exposed to global issues such as addressing climate change. GCAM-v5.2 uses socioeconomic development scenarios in line with the Shared Socio-economic Pathways as drivers to project future demand for each sector. Given the constraints imposed by its inputs (cost, current and future technology, efficiency, resource availability, etc.), GCAM-v5.2 iteratively finds a solution that balances supply and demand across all sectors and minimizes costs. In brief, this model is solved by finding a combination of available technologies and resources with the lowest cost and being most technically feasible. Decision-making in GCAM-v5.2 relies on a logit-choice formulation[71,72], and options with the lowest cost will gain the largest market share, while others will gain a relatively small market share. The detailed model descriptions are summarized in Supplementary Note S1.

### Road transport sector in GCAM

To most effectively simulate the development trend of China's road transportation, we update GCAM-v5.2 to add electric vehicle (EV) and fuel cell vehicle (FCEV) technologies for the bus and truck sectors. Road transportation in GCAM-v5.2 is divided into two sectors, including the passenger sector and the freight sector, which is shown in Fig. S1. The passenger sector can be further divided into two sub-sectors: light duty vehicle-four wheels (LDV-4W) and bus. The LDV-4W subsector includes four different modes—compact car, large car and SUV, mini car, and subcompact car. Among which, each mode has five technology options—internal combustion engine vehicle (ICEV), EV, FCEV, hybrid electric vehicle (HEV), and natural gas vehicle (NGV). Moreover, two modes (light bus and heavy bus) are disaggregated in the bus subsector and three modes (light truck, medium truck, and heavy truck) are involved in the freight subsector, and each mode contains four technology options, including ICEV, EV, FCEV, and NGV. The analysis framework in this study is shown in Supplementary Fig. S2, and the parameters can be found in Supplementary Tables. 1–5. We also summarized the popular method on vehicle flow and stock projection, which is shown in Table S8. Codes needed for this study can be found in Wang, H. et al.[73].

### Material flow analysis

Since GCAM does not count the number of vehicles explicitly, a conversion of transportation service demand into the number of vehicles is required. Eq (13) in Note S5 presents the conversion formula. We

**Table 1 | EV cost change scenarios**

| Scenario name | Future EV cost assumptions |
|---|---|
| Baseline scenario(BLS) | The uptake pace of EVs will fulfil the requirement of China's 2060 carbon neutrality target and the rapid decrease in EV cost will be in line with the historical and forecasted development trend of EV cost in China as reported in the existing literature |
| LIB-High-cost (LIB-High) | A rapid increase in critical material price |
| LIB-Medium-cost (LIB-Medium) | A steady increase in critical material price |
| LIB-Low-cost (LIB-Low) | A slight increase in critical material price which mainly affects EV costs during the middle and later periods of the study |

then adopt a stock-driven dynamic material-flow-analysis (MFA) model to estimate the inflow (sale) and outflow (decommissioning) of vehicles. The technical details are presented in Note S5.

Considering that battery's operational lifetime has a significant impact on material recycling and the EV adoption[74]. We couple a lifetime distribution delay forecasting model with the dynamic MFA to investigate the effects of recycling on EV development, which considers the second use and lifetime of batteries. Please see Note S6 and Fig.S4 for technical details.

## Climate policy scenario
We employ a widely used near-term to net-zero approach to predict China's $CO_2$ emission trajectory under the carbon neutrality target. In this trajectory, $CO_2$ emissions decline linearly from 2020 (3,125 Gt) until they reach net zero by 2060[75,76]. Throughout modeling the net-zero $CO_2$ emission trajectory can to some extent alleviate the uncertainty and measurement difficulty in the reality of the emission pathway estimation, which can also be used to provide a basis for policymaking.

## EV cost change scenarios
We develop four EV cost change scenarios in our analysis (Table 1): base-line scenario (BLS), High-cost (High), Medium-cost (Medium), and Low-cost (Low). (1) In the BLS scenario, the uptake pace of EVs will fulfil the requirement of China's carbon neutrality target by 2060 and the cost of new energy vehicles will continue to decline due to the advancement of low-carbon technology. We assume that EV and ICEV will reach cost parity at around 2025 for LDV-4W, the cost parity between EV and FCEV will arrive during 2025–2030, and afterwards the two cost parities will be maintained. For bus and truck, the cost parity of ICEV and EV will be realized around 2025, while the cost of FCEV will reach parity with EV during 2025–2030, then, the cost decrease of FCEV will become faster than that of EV after 2030[77]. (2) In the High scenario, we assume that the initial strong demand and the resultant supply shortage will drive up the price of critical materials significantly, but in the long run, the increased recycling and use of substitutes will dampen and eventually flatten the price trajectory[78]. (3) The major assumption in the Medium scenario is that a moderate material demand-supply gap will lead to a steady increase in material price, which will ultimately affect the cost of EVs[79]. (4) In the Low scenario, the projection tends to be relatively conservative. Due to the stable growth in both the supply and demand of critical materials in a certain period, the EV cost will remain relatively stable in the early period and will increase slightly in the middle and later periods.

The detailed method for critical material price predictions is summarized in Note S2. The equations linking material price to the costs of LIBs and EVs are displayed in Notes S3-S4. We also compare our results about the LIB cost and EV development under material price surging with existing literature, which is shown in Tables S6-7. The sensitivity analysis of critical material price surging on EV deployment is shown in Fig. S29. The LIB technologies adopted in this study include

nickel cobalt manganese ($LiNi_xCo_yMn_zO_2$) (NCM111, NCM622, NCM811, and NCM9.5.5), nickel cobalt aluminum ($LiCo_{0.15}Al_{0.05}O_2$) (NCA), lithium iron phosphate ($LiFePO_4$) (LFP), and lithium manganese oxide ($LiMn_2O_4$) (LMO). By combining cathode material price change forms with different LIB technologies, we can construct the final EV cost change scenarios, e.g., NCM622-High represents an EV cost change scenario for EVs equipped with an NCM622 LIB, and the EV cost is influenced by the rapid increase in the price of critical materials. The constructed EV cost change scenarios, as well as the projected carbon neutrality trajectory, are inputs into GCAM-v5.2.

## Recycling scenarios
In the recycling scenario (RE), we assume that the recycling is closed-looped, namely, the recycled minerals reaches the quality for battery production[37]. The materials obtained by battery manufacturers through recycling are not affected by material price fluctuations on the international market, that is, only the primary demand for materials is affected by the surging material prices. The material recycling potential is calculated according to the material demand under the BLS scenario.

## Limitations
This research bears several limitations. First, the price surge of materials for EV engines and battery systems, including aluminum, copper, ferrum, etc., and materials used for electricity generation, including neodymium, dysprosium, indium, germanium, argentum, tellurium, etc., could also affect the EV adoption[4,5]. In addition, if the affordability of critical materials use is not guaranteed, carbon emission reductions in both transportation and the related industries would be held back[80]. Further work could expand this analysis to assess how the economic competitiveness of other crucial low-carbon technologies would be affected by the foreseeing surge in the prices of critical materials, how various technologies would compete with each other more strategically, and ultimately how this would influence the realization of climate targets. Second, we do not consider the impact of phosphorus price changes on EV penetration when considering the adoption of LFP batteries given the negligible share of phosphorus in battery cost. However, the surging interest in LFP combined with the rising demand for phosphate from agriculture, the price of phosphorus (and other critical minerals) may move up along a non-stationary path and thus deserves further investigation in future research. Although the price of critical materials is a significant factor affecting the penetration of EVs, we cannot ignore the influence of other factors on the adoption of EVs (e.g., increasing availability and choice in EV models), which should also be paid attention to in future research. Third, it is worth noting that this study does not fully capture price linkages between materials, for example, cobalt prices may decrease with the adoption of low-cobalt batteries and nickel prices may increase accordingly, or lithium prices may increase further with the adoption cobalt-free LFP batteries.

## Data availability
The data generated in this study are provided in Supplementary Information. All data regarding the parameters used in this study are documented in Supplementary Information. More specifically, the mineral price data is shown in Supplementary Fig. S3, where historical prices are available at UGSG (https://www.usgs.gov/centers/national-minerals-information-center/commodity-statistics-and-information) and InfoMine (https://www.mining.com/markets/). All other data (socioeconomic parameters and road transportation sector technology parameters) used in the study are given in Supplementary Tables. S1–S5. Source data are provided with this paper.

## Code availability
The source code for the model is available at https://github.com/JGCRI/gcam-core/releases. Codes for this work can be acquired at https://doi.org/10.5281/zenodo.7652508[73].

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

## Acknowledgements

This research was financially supported by the National Natural Science Foundation of China (72074138 to K.F. and L.S., 72222014 to J.L., 72274187 to P.W., 71961147003 to W.C., 71834004 to K.F.), the Taishan Scholars Program of Shandong Province (to J.L.), Shandong University Interdisciplinary Research and Innovation Team of Young Scholars (to K.F. and J.L.), the Shandong Provincial Science Fund for Excellent Youth Scholars (ZR2021YQ27 to J.L.), and the National Social Science Fund of China (21ZDA065 to J.L., 22VMG017 to J.L.). P. W. acknowledges the support from CAST Young Talent Support Project and CAS Pioneer Hundred Talents Program.

## Author contributions

H.W., K.F., P.W., L.S., and J.L. designed the research. H.W., K.F., P.W., Y.Y., L.S., F.Y., and J.L. conducted the analysis. H.W., K.F., P.W., L.S., W.C., Y.Z., and J.L. led the drafting of the manuscript. All authors contributed significantly to the final writing of the article.

## Competing interests

The authors declare no competing interests.
