## [Peer review file · Nature Communications]

REVIEWER COMMENTS

Reviewer #1 (Remarks to the Author):

This paper analyzes the prices of battery materials of China's electric vehicles over time. The topic is important for policy makers to introduce the EVs in a society. My primary concern is the estimation framework and its novelty. I understand that the equation (13) of Note S5 is crucial in this study. Eq.(13) says that the energy consumption for vehicles causes the number of vehicle stock. Then, the vehicle flow is determined under the given vehicle stock. The framework is strange. The vehicle stock should be survived following their lifespan. Then, the vehicle flow of each type should be determined by consumers. I see that the modern vehicle flow and stock analysis framework has been rapidly developed in relevant journals such as Journal of Industrial Ecology. I am wondering the state of the art. In addition, I am also wondering the lifespan of the EVs purchased in China. Due to a shorter lifespan, the replacement cycle of EVs in China may be fast. This can induce many EVs in secondhand car market and scrap car market in China. The markets can contribute to recycling battery materials in the future. In analyzing it, a more sophisticated analysis framework is necessary. I would suggest a careful work considering the state of the art.

Reviewer #2 (Remarks to the Author):

This paper provides a valuable contribution to the literature in showcasing how the impact of the pricing of key critical materials has the potential to impact consumer uptake of EVs. The work is noteworthy and relevant and I think that it makes an important contribution to the field.

Lines 28-30 Page 1

I am a passionate advocate of EV battery recycling. That said, there is a temporal nature to when materials from EV recycling will be available for use.

Page 2 Lines 30-35

Yes, the price did surge after initial market shocks but then stabilised. There is also another narrative around LME cancelling trades, and how traders felt that this disproportionately benefited some investors over others. In particular, some pointed to the Hong Kong ownership of the LME and the impression that cancelling trades benefited Chinese traders. Of course others have countered this. This has been covered in a range of Financial News articles. It may be worth expanding upon this to showcase the full story as this is especially relevant given the focus on the Chinese market.

Page 15 Line 403.

Given the surge in interest in Lithium Ferrophosphate batteries, is it worth also considering Phosphate Rock in this analysis? This offers an alternative to many of the battery chemistries that are more intensive in their use of more highly critical materials, also there is an interesting sidebar here re: competition with Agriculture for Phosphate e.t.c. The EU added Phosphate Rock to its Critical Materials list in 2020.

Supplementary Information Page 3 Line 55

I couldn't see it, but does the model take into account the concomitant drop in the prices of fossil fuels. There may be a number of factors that affect this and the effects may be distributed differently in different geographies. The West's sanctions on Russia have affected fossil fuel prices, however China's lack of sanctions may mean China has access to cheap energy from Russia. Furthermore, as EVs displace ICEVs, presumably demand for petroleum products will begin to slow and in the absence of additional levies / taxes, there will presumably be a change in the demand - supply balance. This will further exacerbate the life-cycle cost differential between ICEVs and EVs.

Page 11 Line 280

I see the section about the ongoing geopolitical tensions with Russia / Ukraine and the effect that this may have on Critical Raw Materials. Given the Chinese context, perhaps it is worth saying that given China's stance on the issue and lack of any sanctions, China may in effect be a net beneficiary of this situation, as companies in the West cancel trades with Russian metals firms, Nornickel e.t.c.

Page 12 308 - 313

It is perhaps worth explaining that recycled content from manufacturing scrap is available relatively quickly, however recycled material from end of life batteries is likely to take some time to return into the cycle, and so may not be available for some time.

I suppose that there is also an implication here, that our patterns of consumption of private mobility do not change. It may be worth a comment, that given the constraints around Critical Material sourcing, other policy measures may need to be taken to increase the intensity with which we make use of extracted resources. Social fixes like product-service systems and the "uberisation" of vehicles, may allow us to serve more users using less vehicles in a resource constrained scenario, as private vehicles are a poorly optimised asset spending most of their time parked. Perhaps this study points to the need for unconventional solutions and public policy interventions as business as usual with ICEVs cannot continue.

Page 16 Line 413 On

I understand the limitations on many other materials. I'd perhaps question why Phosphate Rock isn't amongst the materials under evaluation given its prominence in LFP which is likely to become an increasingly dominant cathode chemistry.

My real query here is that my understanding is that changes in ICEV prices are modelled by on the flip side the total-cost of ownership of ICEVs is not. If consumers are making a choice between competing technologies, is it assumed that the prices of one ICEV stays relatively constant? I am not sure if e.g. fuel becomes cheaper if more pivot to EVs as there is less demand for Hydrocarbon fuels in transportation, or whether oil cartels will crimp output accordingly to maintain prices? Also... when we get to the point where EV vehicles are dominant, I wonder if the costs of maintaining the infrastructures for fossil fuels gets spread across a dwindling pool of consumers. I think for balance, the paper needs a section about how the total cost of ownership of ICEV vehicles will evolve in the transition.

Reviewer #3 (Remarks to the Author):

Dear Laixiang Sun,

Thank you for the opportunity to review this paper. I found the analysis to be timely and helpful in framing the discussion of electric vehicle adoption in the face of potentially higher material costs.

My general comments and suggestions are as follows:

All the charts in Figure 4 are labeled as -high; I think these are supposed to be "high", "medium" and "low"?

NMC111 is used as the example chemistry for discussion. I would suggest using NMC622 which is much more common for EVs now.

Prices for materials are likely to be linked (i.e., as low cobalt batteries are adopted, the price of cobalt may decrease and the price on nickel may correspondingly increase). This is difficult to capture in the methodology used in the paper, but should be discussed in the limitations section.

The analysis also does not account for the effect of increasing availability and choice in electric vehicle models, which is likely to impact adoption independent of price. Another limitation of the study.

Although recycling is mentioned in the abstract and discussion, it is not discussed quantitatively in the paper. What is the percentage decrease in prices assumed to be attributable to recycling in the 3 scenarios?

The analysis is likely to be sensitive to starting assumptions (e.g., the nominal value in the medium price scenario). The supplementary information should include a more in-depth discussion of how these values were derived and their impact on the final results.

Response to Reviewers' Comments on Manuscript NCOMMS-22-29602-T

Title: Surging Critical Material Prices Jeopardize China's Electric Vehicle and Climate Ambitions

Reviewer #1:

This paper analyzes the prices of battery materials of China's electric vehicles over time. The topic is important for policy makers to introduce the EVs in a society. My primary concern is the estimation framework and its novelty. I understand that the equation (13) of Note S5 is crucial in this study. Eq.(13) says that the energy consumption for vehicles causes the number of vehicle stock. Then, the vehicle flow is determined under the given vehicle stock. The framework is strange. The vehicle stock should be survived following their lifespan. Then, the vehicle flow of each type should be determined by consumers. I see that the modern vehicle flow and stock analysis framework has been rapidly developed in relevant journals such as Journal of Industrial Ecology. I am wondering the state of the art. In addition, I am also wondering the lifespan of the EVs purchased in China. Due to a shorter lifespan, the replacement cycle of EVs in China may be fast. This can induce many EVs in secondhand car market and scrap car market in China. The markets can contribute to recycling battery materials in the future. In analyzing it, a more sophisticated analysis framework is necessary. I would suggest a careful work considering the state of the art.

Response: Thank you very much for your suggestions.

- (1) Sorry for the confusion caused by Eq. (13), which has been re-formulated in the revision. In our analysis, the estimation of future vehicle stocks is determined by the consumers' purchase demand rather than energy consumption. Our study used GCAM to project vehicle stock in terms of the transportation services (in passenger-km, tonne-km), which are ultimately driven by population, GDP, and aggregate service price level. Transportation services consume energy to produce outputs of passenger-km and tonne-km. Therefore, we used energy consumption as the proxy for vehicle stock calculation in the original version. From your comments we recognize that this may cause confusion. To avoid the confusion, we reformulate Eq. (13) as follows:

$$Veh = D \times L^{-1} \times VKT^{-1} \quad (13)$$

where Veh stands for the vehicle stock; D is the transportation demand (passenger-km or tonne-km); L is the load factor (persons or tonnes) per vehicle; VKT is the vehicle travelled kilometer (km/vehicle).

We add Fig. S2 in the SI to show the schematic diagram of the calculation principles in GCAM. We have also revised Notes S1 and S5 in the SI to provide the detailed calculation principles regarding the vehicle stocks in GCAM.

- (2) Following the above suggestion, we have further investigated the state of the art in vehicle flow and stock projection, and summarized this in-depth review in Table S8. At present, global transportation models, mainly including Global Change Assessment Model (GCAM)¹, MESSAGE-Transport (Model for Energy Supply Strategy Alternatives and their General Environmental Impact)², MoMo (Mobility Model)³, and Roadmap⁴, are widely used to simulate transport development. By considering the linkages with global land use, energy/economic, and/or climate systems, GCAM and MESSAGE tend to rely on cross-sectoral endogenous functions (population and income (GDP)) to project future vehicle development, whereas MoMo and Roadmap rely more heavily on expert judgment and detailed, country-

specific research and expertise. Some other driving factors such as income distribution and vehicle price variation have also been used by some models for projecting future vehicle stocks⁵. Among these methods, GCAM can incorporate general representations of the whole energy systems and various technology options into a consistent framework^{6,7}. In more detail, GCASM-v5.2 has the advantage of modeling the endogenous interactions of transportation with other sectors within an individual region, as well as with other regions, and therefore, this integrated framework is well-suited for analyzing China's transportation development when exposed to global issues such as addressing climate change.

Our critical review of the literature on vehicle stock and flow analysis reveals that the previous studies overlook the role of critical materials in determining the future cost dynamics of EVs. Future surge in the prices of critical materials may significantly undermine the competitiveness of EVs. A negligence of such an important cost factor may lead to a biased estimation of China's EV development in the future, which would in turn affect the delivery of China's carbon neutral commitment. To address this important gap we extended the GCAM to capture the additional costs for EVs. The extended model is able to analyze the effects of metal price changes on EV adoption. Meanwhile, we also update GCAM v5.2 to add EV and FCEV technologies for the bus and truck sectors so as to improve the projections on the development trend of China's road transport sector (please see Notes S2, S3, and S4).

To improve our modelling framework as you suggested, we develop a new model which couples the lifetime distribution delay forecasting model with the material flow analysis. The new modelling setting is able to cover the lifecycle stages of an EV battery, including repaired, reused, remanufactured, second use, direct recycling, etc., to capture the impact of battery operational life on critical battery material recycling. The detailed description of the method is presented in Note S6 and Fig. S4.

The upgraded framework in the revision consists of the extended GCAM, stock-driven dynamic material-flow-analysis (MFA) model, and lifetime distribution delay forecasting model. The implementation procedures are presented in Fig. S2, which include three major steps. We first forecast and update the EV cost which is in line with the historical and forecasted development trend of EV cost in China as reported in the existing literature (Table S5). On this basis, we include metal price change into the battery production price trajectory, to capture the additional costs to EVs, which will serve as an input for vehicle stock projection in GCAM (the method is described in Notes S2, S3, and S4). Secondly, we run the GCAM analysis and material flow analysis (Notes S1 and S5) to generate results related to EV penetration rate and the corresponding carbon emissions under material price surging scenarios. Thirdly, we run the lifetime distribution delay forecasting model and material flow analysis (Note S6) to investigate the material recycling potential. Material recycling affects the EV cost by reducing the primary demand for these critical materials which is subject to volatilities on international market. These new EV cost trajectories can be feedback to GCAM as new inputs to reveal the impact of material recycling on EV development. Our findings demonstrate that the price surge of critical materials will jeopardize the fleet electrification and put additional pressure on China's carbon neutral ambition, while material recycling of LIBs is promising in addressing the material price challenge, especially in the long-term. We present the results and discussions on the effects of material recycling in the revised manuscript (lines 281-336, 371-402).

Figure S2. Schematic diagram of our analysis framework. *Note:* VKT is vehicle kilometers travelled whereas PKT is passenger kilometers travelled (PKT is related to VKT through the number of passengers per vehicle, which is sometimes called the occupancy rate).

Table S8. Literature review on vehicle flow and stock projection

Author	Method	Key results
Milovanoff et al. (2020) ⁸	GCAM: The demand for passenger transportation services depends on per-capita GDP, the aggregated service price across all modes, the population, and income and price elasticities. Then, the market shares by mode and technology are determined using a logit formulation based on the cost of transport service and other cost parameters.	Current US policies are insufficient to remain within a sectoral CO ₂ emission budget for light-duty vehicles, consistent with preventing more than 2 °C global warming, creating a mitigation gap of up to 19 GtCO ₂ (28% of the projected 2015–2050 light-duty vehicle fleet emissions). Closing the mitigation gap solely with EVs would require more than 350 million on-road EVs (90% of the fleet), half of national electricity demand and excessive amounts of critical materials to be deployed in 2050.
McCollum et al. (2018) ⁹	Six global energy economy modelling frameworks were employed in this study: GEM-E3T-ICCS, IMACLIM-R, IMAGE, MESSAGE-Transport, TIAM-UCL and WITCH. (1) GEM-E3T-ICCS: The stock of vehicles by transport sector and the cars, represented as durable goods in the modelling of behavior of households, change over time as a result of	A diverse set of measures targeting vehicle buyers is necessary to drive widespread adoption of clean technologies. Carbon pricing alone is insufficient to bring low-carbon vehicles to the mass market, though it may have a supporting role in

	mobility and scrappage. The choice of between vehicle technologies depends on relative costs, which include purchasing cost, running costs and cost factors reflecting uncertainty factors. (2) IMACLIM-R: The service demand is determined by demography and labor productivity growth, the maximum potentials of technologies, the learning rates decreasing the cost of technologies, fossil fuel reserves, the parameters of the functions representing energy-efficiency in end-uses, and the parameters of the functions representing energy-demand behaviors and life-styles. (3) IMAGE: The service demand is determined by GDP and population projections. (4) MESSAGE-Transport: Future demand for passenger travel in the various modes is projected on a passenger-kilometer (pkm) basis as a function of per-capita GDP. (5) TIAM-UCL: The service demands projected are calculated from a set of exogenously defined drivers (e.g., GDP, population, number of households); the demands respond to prices. (6) WITCH: Transport demand is explicitly calculated based on GDP and population projections.	ensuring a decarbonized energy supply.
Isik et al. (2021) ¹⁰	COMET model: Transport demands are derived by gross domestic product, population, etc.	The electrification of light-duty vehicles at earlier periods is essential for deeper reductions in air emissions. When further combined with energy efficiency improvements, these actions contribute to CO₂ reductions under the scenarios of more CO₂-intense electricity.
Baars et al. (2021) ¹¹	Ricardo Sultan model: Projections for future car sales are based on the average car ownership per 1000 inhabitants in 2017, multiplied by future population projections.	The rapid development of EVs will lead to widespread adoption of LIBs, which will require increased natural resources for the automotive industry. The expected rapid increase in batteries could result in new resource challenges and supply-chain risks.
Hao et al. (2019) ¹²	Transport Impact Model (TIM): (1) Private passenger vehicle growth model: Automotive growth is correlated with household income growth and vehicle price variation. (2) Urban public transportation vehicle growth model: Automotive growth is correlated with urbanization and population growth. (3) Economic utility vehicle growth model: Automotive growth is correlated with GDP growth.	A mass electrification of the heavy-duty segment on top of the light-duty segment would substantially increase the lithium demand and impose further strain on the global lithium supply.

Peng et al. (2018) ¹³	China Provincial Road Transport Energy Demand and GHG Emissions Analysis (CPREG) model: (1) Non-taxi passenger vehicle stocks are projected with the Gompertz function relating vehicle ownership to per-capita GDP. (2) The stock of commercial buses is the product of the ownership and population. (3) Freight vehicles stock is assumed to be correlated to the elasticity between vehicle stock and GDP.	China's vehicle stock will keep increasing to 543 million by 2050. The spatial distributions of future vehicle stock, energy demand and GHG emissions vary among provinces and show a generally downward trend from east to west.
Pan et al. (2018) ¹⁴	GCAM-TU: (1) Passenger demand is determined by income (per capita GDP), population, and aggregate service price. (2) Freight demand trajectory is estimated based on population and GDP that is subject to price-induced demand response.	China's transportation sector might need significant changes beyond 2030 to decouple associated CO ₂ emissions from GDP growths. Supporting national mitigation has more pronounced implications on freight than passenger transport services, and arouses a radical shift of transport fuels away from fossil-based liquids to clean alternatives.
Khanna et al. (2021) ¹⁵	Demand Resource Energy Analysis Model (DREAM): The future sales and the implied stock of heavy-duty trucks is estimated by a bottom-up stock turnover model.	Beginning to deploy battery electric and fuel-cell heavy-duty trucks (HDTs) as early as 2020 and 2035, respectively, could achieve significant and the largest CO ₂ emissions reduction by 2050 with a decarbonized power sector.

Pages 12-13, Line 281-336 in Main Text:

Material Recycling Promotes Fleet Electrification

Fig. 5 report the results under the combinations of the RE (recycling) scenario and the High, Medium, and Low scenarios. The recycling potential of materials shows an increasing upward trend (Fig. 5a). Due to the delayed effects of material recycling, the resulting proportion of recycled materials to the total material demand in the LDV-4W, bus, and truck sector will be only 3%, 18%, and 3%, respectively, in 2030, however, this value could reach 85%, 86%, and 70%, respectively, by 2060. The recycled materials reduce the extent to which the materials needed for EVs are exposed to material price surges on international markets, thus reducing the likelihood of cost surging for EVs. Taking EVs equipped with NCM622 LIBs as an example (Figs. 5a and S33), the EV cost will decrease to about 0.05-0.15 (1990)\$/pass-km by 2030 for LDV-4W under High-RE, Medium-RE, and Low-RE scenarios, which are slightly lower than those in High, Medium, and Low scenarios, respectively. In the bus and truck sector, material recycling will only help decrease the EV cost by about 1% by 2030. But the benefits of material recycling can be significant in the long-term. By 2060, the cost of EVs in the High-RE, Medium-RE, and Low-RE scenarios will be 11-15%, 4-6% and 4-5% lower than those in the High, Medium, and Low scenarios in the LDV-4W sector to reach about 0.05-0.13 (1990)\$/pass-km, which is basically the same as that in the BLS scenario (even in the High scenario). The EV cost in bus and truck sector will have about 2-10% decrease in RE scenarios by 2060. These results clearly manifest that material recycling can greatly reduce the impact

of surging material prices on the EV cost, especially in the long-term.

The decrease in EV cost will raise EV penetration rate. As shown in Fig. 5b, the material recycling will help increase the EV (with NCM622 LIBs) penetration rate by 7, 1, and 1 percentage points by 2030 in High-RE, Medium-RE, and Low-RE scenarios, respectively, and the resultant rates are still 14%, 8%, and 6% lower than those in BLS scenario. However, the recycling will boost the EV uptake rate to 59%, 66%, and 67% in High-RE, Medium-RE, and Low-RE scenarios, respectively, by 2060 (much closer to 67% under the BLS). This will inevitably reduce ICEV's market share by 12 percentage points (to about 10%), 2 percentage points (to about 6%), and 2 percentage points (to about 6%), making ICEV's market share close to that in the BLS (6%). Let all EVs be equipped with cobalt-free LIBs, recycling can reduce ICEV market penetration to baseline levels even in the High-RE scenario (other RE-combining scenarios than NCM622-RE can be found in Figs. S39-45). These results highlight that recycling would have remarkable effects to mitigate the material price challenge in EV development in the long-term.

Due to the positive role of material recycling in promoting EV development, the resulting cumulative CO₂ emissions from road transportation in 2020 to 2060 under High-RE, Medium-RE, and Low-RE scenarios decrease to 22 Gt, 20 Gt, and 19 Gt, respectively, which are 8%, 2%, and 1% lower than those under High, Medium, and Low scenarios, respectively (17%, 9%, and 4% higher than those in BLS scenario) (Fig. 5c). Although the CO₂ emissions from road transportation in the RE-combining scenarios will only decrease by less than 1% compared with that in High, Medium, and Low scenarios by 2030. After 2030, the differences in CO₂ emissions between the RE and BLS scenarios become narrowing. By 2060, the CO₂ emissions will decrease to 0.27 Gt/yr, 0.24 Gt/yr, and 0.23 Gt/yr in High-RE, Medium-RE, and Low-RE scenarios, respectively, which are 36%, 12%, and 10% lower than those in High, Medium, and Low scenarios (Fig. 5d). For EVs with cobalt-free LIBs, materials recycling can reduce the cumulative CO₂ emissions to a level only 2-7% higher than those under the BLS (Fig. S46). This indicates that materials recycling can facilitate low-carbon transition in the transportation sector in the long-term.

Fig. 5. Material recycling effects on EV development and CO₂ emissions of road transportation in China from 2020 to 2060 under the RE scenario: a) Material recycling potential and the effect on EV cost evolution; b) EV and ICEV penetration rate, c) cumulative CO₂ emissions, d) CO₂ emissions by year. *Note:* The scenarios of **BLS**, **High**, **Medium**, and **Low** are the same as in Fig. 1; RE is a scenario in which only primary demand is affected by the market price of the material concerned; MS is the market share of vehicles; All EVs in this figure are equipped with NCM622 LIBs.

Page 19, Lines 493-502 in **Main Text**:

Material flow analysis

Since GCAM does not count the number of vehicles explicitly, a conversion of transportation service demand into the number of vehicles is required. Eq (13) in Note S5 presents the conversion formula. We then adopt a stock-driven dynamic material-flow-analysis (MFA) model to estimate the inflow (sale) and outflow (decommissioning) of vehicles. The technical details are presented in Note S5.

Considering that battery's operational lifetime has a significant impact on material recycling and the EV adoption⁷⁴. We couple a lifetime distribution delay forecasting model with the dynamic MFA to investigate the effects of recycling on EV development, which considers the second use and lifetime of batteries. Please see Note S6 for technical details.

Recycling scenarios

In the recycling scenario (RE), we assume that the recycling is closed-looped, namely, the recycled minerals reaches the quality for battery production³⁷. The materials obtained by battery manufacturers through recycling are not affected by material price fluctuations on the international market, that is, only the primary demand for materials is affected by the surging material prices. The material recycling potential is calculated according to the material demand under the BLS scenario.

Notes S5 and S6 in Supplementary Information:

Note S5. The vehicle stock and sale calculation

Since GCAM does not count the number of vehicles explicitly, a conversion of transportation service demand into the number of vehicles is required. The vehicle stock can be calculated using the following equation:

$$Veh = D \times L^{-1} \times VKT^{-1} \quad (13)$$

where Veh stands for the vehicle stock; D is the transportation demand (passenger-km or tonne-km); L is the load factor (persons or tonnes) per vehicle; VKT is the vehicle travelled kilometer (km/vehicle).

We then adopt a stock-driven dynamic material-flow-analysis (MFA) model to estimate the inflow (sale) and outflow (decommissioning) of vehicles.

$$\text{outflow}_{EV}(t_n) = \sum_{t_0}^{t_n-1} (\text{inflow}_{EV}(t_i) \cdot [\text{survival}(t_{i-1} - t_0) - \text{survival}(t_i - t_0)]) \quad (14)$$

Where the outflow in t_n ($\text{outflow}_{EV}(t_n)$) is the sum of decommissioning of past inflow vintages in $t_i \in (t_0, t_{n-1})$. Survival(Δt) is the complementary cumulative distribution function of the normal distribution¹⁰. In this study, the average lifetime is assumed to be 10 years, 13 years, and 15 years for LDV-4W, bus, and truck, respectively. According to the principle of conservation of mass, the inflow ($\text{inflow}_{EV}(t_n)$) must equal a combination of the changes in stock ($\text{stock}_{EV}(t_n) - \text{stock}_{EV}(t_{n-1})$) and all outflows during this period:

$$\text{inflow}_{EV}(t_n) = \text{stock}_{EV}(t_n) - \text{stock}_{EV}(t_{n-1}) + \text{outflow}_{EV}(t_n) \quad (15)$$

Note S6. The effects of second use and battery lifetime on material recycling

Closed-loop recycling of battery materials is an important source of future battery material supply¹¹, however, the changes in battery operation lifetime will have a significant impact on material recycling. Therefore, we couple the lifetime distribution delay forecasting model with the material flow analysis to analyze the recycling potential of battery materials by considering both direct and indirect battery returns from first use and second use¹². We assume that the materials obtained by battery manufacturers through recycling are not affected by material price fluctuations on the international market, that is, only the primary demand for materials is affected by the surging material prices.

The lifecycle stages of an EV battery and the sources of waste batteries entering the recycling market are illustrated in Fig. S4. EV waste batteries entering the recycling market include direct and indirect sources. Indirect sources include decommissioned batteries after secondary applications (such as energy storage systems) and batteries replaced by early failures (replaced batteries can only be recycled). Batteries that do not belong to the early

failure and cannot be echelon utilized and can only be recycled belong to the direct source. Here, early failure refers to a failure that occurs during the warranty period of an EV (8 years), some of the battery faults () can be repaired, reused, or remanufactured again for use in EVs, and other part of faulty batteries (1 -) can only be replaced, and the replaced batteries are recycled. The early failure remanufactured rate is assumed to be 70% in this research.

The amounts of replacements, $QR(w)$ at year w for batteries sold in year s is calculated using Eq. (16).

$$QR(w) = \sum_{s=w-8}^{s=w-1} QP(s) \times D_{EV}(s, w) \times (1 - r(s)) \quad (16)$$

Where $QP(s)$ is the total number of EVs put on the market in year s in the BLS scenario; $D_{EV}(s, w)$ is the probability of product failure in year w of a battery that started its use stage in year s ; The average lifetime of the use stage is set to be 11 years, and the standard deviation is set as 1.8.

The direct waste batteries that fail outside the warranty period and directly flow onto the recycling market in year w ($DW(w)$) can be estimated with the following equation:

$$DW(w) = \sum_{s=1}^{s=w-9} (QP(s) + QR(s)) \times D_{EV}(s, w) \times S(w) \quad (17)$$

Where $S(w)$ is the share of recycling EV battery on EoL markets, which will be reduced from 90% in 2019 to 50% in 2030 and stay stable afterwards.

The amount of waste EV batteries flowing into the B2U application (Q_{B2U}) can be calculated by eq. (18):

$$Q_{B2U} = \sum_{s=1}^{s=w-9} (QP(s) + QR(s)) \times D_{EV}(s, w) \times (1 - S(w)) \quad (18)$$

Where $1-S(w)$ is the share of second use EV battery on EoL markets.

The composition of retired EV batteries from B2U applications $IW_{B2U}(w)$, is formulated in Eq. (19).

$$IW_{B2U}(w) = \sum_{s=1}^{s=w-1} Q_{B2U}(s) \times D_{B2U}(s, w) \quad (19)$$

Where $D_{B2U}(s, w)$ is the probability of product failure in year w of a battery that started its use stage in year s . The average lifetime of the lifetime of EV batteries in B2U is set to be 5 years, and the standard deviation is set as 2.6.

Thus, the total waste stream returning to recycling in year w ($TW(w)$), termed as the sum of direct recycled batteries (DW), waste batteries from second use applications (IW_{B2U}), and replacement EV batteries (QR), is given by Eq. (20).

$$TW(w) = DW(w) + IW_{B2U}(w) + QR(w) \quad (20)$$

The amount of recycling material in year w ($FC_{recycling}(w)$) can be estimated with Eq. (21):

$$FC_{recycling}(w) = TW(w) \times Cap_{EV}(s) \times C_i(s) \quad (21)$$

Where $Cap_{EV}(s)$ is their average capacity in kWh in year s , which is 35 kWh for LDV-4W, 70 kWh for bus, and 106 kWh for truck; $C_i(s)$ is material intensity of material i (kg/kWh) in EV batteries sold in year s , which is summarized in Table S2.

Figure S4 in Supplementary Information:

Figure S4. Sources of waste batteries entering the recycling market. *Note:* r is the share of early failures of batteries during the warranty period that can be remanufactured; S is the share of recycling in EoL markets; QR is the amount of replacement EV batteries; DW is the amount of direct waste EV batteries; IW_{B2U} is the amount of waste EV batteries after second use applications.

Reviewer #2:

This paper provides a valuable contribution to the literature in showcasing how the impact of the pricing of key critical materials has the potential to impact consumer uptake of EVs. The work is noteworthy and relevant and I think that it makes an important contribution to the field.

Lines 28-30 Page 1. I am a passionate advocate of EV battery recycling. That said, there is a temporal nature to when materials from EV recycling will be available for use.

Response:

Thank you for your suggestions. We agree that there is a temporal nature in terms of when materials from EV recycling will be available for use. To address this concern, we couple a lifetime distribution delay forecasting model with the material flow analysis to investigate the effects of material recycling on EV development in the context of material price surge and considering the second use and lifetime of batteries. This extension of the study confirms the importance of EV battery recycling in promoting sustainable transportation development under the condition of material scarcity constraint, suggesting that there will be about 10 years to witness the booming material recycling after 2030. Therefore, we have reason to believe that the recycling of LIBs will be timely in addressing the long-term material challenge.

Accordingly, we add the relevant results and discussions:

In the main text:

Page 1 (Abstract), Lines 28-30:

Material recycling and technical innovation of lithium-ion batteries (LIBs) are effective countermeasures to cope with the material cost surging challenge, especially in the long-term.

Pages 12-13, Line 281-336:

Material Recycling Promotes Fleet Electrification

Fig. 5 report the results under the combinations of the RE (recycling) scenario and the High, Medium, and Low scenarios. The recycling potential of materials shows an increasing upward trend (Fig. 5a). Due to the delayed effects of material recycling, the resulting proportion of recycled materials to the total material demand in the LDV-4W, bus, and truck sector will be only 3%, 18%, and 3%, respectively, in 2030, however, this value could reach 85%, 86%, and 70%, respectively, by 2060. The recycled materials reduce the extent to which the materials needed for EVs are exposed to material price surges on international markets, thus reducing the likelihood of cost surging for EVs. Taking EVs equipped with NCM622 LIBs as an example (Figs. 5a and S33), the EV cost will decrease to about 0.05-0.15 (1990)\$/pass-km by 2030 for LDV-4W under High-RE, Medium-RE, and Low-RE scenarios, which are slightly lower than those in High, Medium, and Low scenarios, respectively. In the bus and truck sector, material recycling will only help decrease the EV cost by about 1% by 2030. But the benefits of material recycling can be significant in the long-term. By 2060, the cost of EVs in the High-RE, Medium-RE, and Low-RE scenarios will be 11-15%, 4-6% and 4-5% lower than those in the High, Medium, and Low scenarios in the LDV-4W sector to reach about 0.05-0.13 (1990)\$/pass-km, which is basically the same as that in the BLS scenario (even in the High scenario). The EV cost in bus and truck sector will have about 2-10% decrease in RE scenarios by 2060. These results clearly manifest that material recycling can greatly reduce the impact of surging material prices on the EV cost, especially in the long-term.

The decrease in EV cost will raise EV penetration rate. As shown in Fig. 5b, the material recycling will help increase the EV (with NCM622 LIBs) penetration rate by 7, 1, and 1 percentage points by 2030 in High-RE, Medium-RE, and Low-RE scenarios, respectively, and the resultant rates are still 14%, 8%, and 6% lower than those in BLS scenario. However, the recycling will boost the EV uptake rate to 59%, 66%, and 67% in High-RE, Medium-RE, and Low-RE scenarios, respectively, by 2060 (much closer to 67% under the BLS). This will inevitably reduce ICEV's market share by 12 percentage points (to about 10%), 2 percentage points (to about 6%), and 2 percentage points (to about 6%), making ICEV's market share close to that in the BLS (6%). Let all EVs be equipped with cobalt-free LIBs, recycling can reduce ICEV market penetration to baseline levels even in the High-RE scenario (other RE-combining scenarios than NCM622-RE can be found in Figs. S39-45). These results highlight that recycling would have remarkable effects to mitigate the material price challenge in EV development in the long-term.

Due to the positive role of material recycling in promoting EV development, the resulting cumulative CO₂ emissions from road transportation in 2020 to 2060 under High-RE, Medium-RE, and Low-RE scenarios decrease to 22 Gt, 20 Gt, and 19 Gt, respectively, which are 8%, 2%, and 1% lower than those under High, Medium, and Low scenarios, respectively (17%, 9%, and 4% higher than those in BLS scenario) (Fig. 5c). Although the CO₂ emissions from road transportation in the RE-combining scenarios will only decrease by less than 1% compared with that in High, Medium, and Low scenarios by 2030. After 2030, the differences in CO₂ emissions between the RE and BLS scenarios become narrowing. By 2060, the CO₂ emissions will decrease to 0.27 Gt/yr, 0.24 Gt/yr, and 0.23 Gt/yr in High-RE, Medium-RE, and Low-RE scenarios, respectively, which are 36%, 12%, and 10% lower than those in High, Medium, and Low scenarios (Fig. 5d). For EVs with cobalt-free LIBs, materials recycling can reduce the cumulative CO₂ emissions to a level only 2-7% higher than those under the BLS (Fig. S46). This indicates that materials recycling can facilitate low-carbon transition in the transportation sector in the long-term.

Fig. 5. Material recycling effects on EV development and CO₂ emissions of road transportation in China from 2020 to 2060 under the RE scenario: a) Material recycling potential and the effect on EV cost evolution; b) EV and ICEV penetration rate, c) cumulative CO₂ emissions, d) CO₂ emissions by year. *Note:* The scenarios of **BLS**, **High**, **Medium**, and **Low** are the same as in Fig. 1; RE is a scenario in which only primary demand is affected by the market price of the material concerned; MS is the market share of vehicles; All EVs in this figure are equipped with NCM622 LIBs.

Page 19, Lines 493-502:

Material flow analysis

Since GCAM does not count the number of vehicles explicitly, a conversion of transportation service demand into the number of vehicles is required. Eq (13) in Note S5 presents the conversion formula. We then adopt a stock-driven dynamic material-flow-analysis (MFA) model to estimate the inflow (sale) and outflow (decommissioning) of vehicles. The technical details are presented in Note S5.

Considering that battery's operational lifetime has a significant impact on material recycling and the EV adoption⁷⁴. We couple a lifetime distribution delay forecasting model with the dynamic MFA to investigate the effects of recycling on EV development, which considers the second use and lifetime of batteries. Please see Note S6 for technical details.

Page 20, Lines 544-550:

Recycling scenarios

In the recycling scenario (RE), we assume that the recycling is closed-looped, namely, the recycled minerals reaches the quality for battery production³⁷. The materials obtained by battery manufacturers through recycling are not affected by material price fluctuations on the international market, that is, only the primary demand for materials is affected by the surging material prices. The material recycling potential is calculated according to the material demand under the BLS scenario.

Page 2 Lines 30-35. Yes, the price did surge after initial market shocks but then stabilised. There is also another narrative around LME cancelling trades, and how traders felt that this disproportionately benefited some investors over others. In particular, some pointed to the Hong Kong ownership of the LME and the impression that cancelling trades benefited Chinese traders. Of course, others have countered this. This has been covered in a range of Financial News articles. It may be worth expanding upon this to showcase the full story as this is especially relevant given the focus on the Chinese market.

Response:

Thank you for your suggestions. Indeed, the price of metal is highly volatile. As mentioned in the introduction, this research focuses on the impact of long-term price movement, rather than short-term volatilities, on the uptake of EVs. The comment regarding the LME is quite interesting. There is indeed disagreement about the LME's cancellation of nickel trade. The LME has enraged some of the world's most influential electronic traders after it shut down its nickel market and unwound thousands of deals in response to a spike in the price of the metal. For example, AQR, one of the largest hedge funds in the world, is exploring legal options in its dispute with the LME after losing significant amount of money following the exchange's decision. However, LME emphasized that its decision had taken due regulatory process into account and was in the interest of the market as a whole. The sharp fluctuation of nickel price has a great impact on the spot industry. Many enterprises that imported raw materials based on the current month's exchange price said that they can no longer bear the high losses, and the industrial chain has been affected by the short-term rapid rise in costs, so they cannot reasonably price for a long time. From this point of view, it was reasonable for LME to cancel the trading on the same day. At that time, the nickel price was already out of the spot reality, and if this situation was allowed to develop unchecked, the credibility of the exchange's pricing mechanism would be affected, and even a series of defaults might be triggered. Accordingly, we have extended the discussion with necessary details in the introduction (lines 39-48).

In the main text:

Page 2 Line 39-48:

For example, nickel price has been very volatile in 2022. On 8 March 2022, it topped \$100,000 per ton before the London Metal Exchange (LME) was forced to step in and halt trading for the next few days, which "has never happened before in the history of the nickel market"⁶. Some hedge funds argued that the LME's decision constituted an injury to their own rights and interests, and they wanted to seek compensation. However, LME emphasized that its decision had taken due regulatory process into account and was in the interest of the market as a whole. Although nickel price has retreated from this peak, it is still relatively high. This

type of volatility not only makes the market trend difficult to predict, but also puts great pressure on the EV market which depends on lithium-ion battery (LIB).

Page 15 Line 403. Given the surge in interest in Lithium Ferrophosphate batteries, is it worth also considering Phosphate Rock in this analysis? This offers an alternative to many of the battery chemistries that are more intensive in their use of more highly critical materials, also there is an interesting sidebar here re: competition with Agriculture for Phosphate e.t.c. The EU added Phosphate Rock to its Critical Materials list in 2020.

Response:

The changes in the cost of phosphorus is not considered in this study because the cost of phosphorus accounts for less than 2% of the LFP cost¹⁷ and the price of phosphorus is much lower and less volatile compared to the studied critical materials. To be more specific, the cathode accounts for about 17% of the LFP cost, of which iron phosphate accounting for about 20% of cathode cost (with phosphorus accounting for about 53% of iron phosphate cost). For the same reason, this study neglects changes in the cost of any “other” cathode active materials such as aluminum in NCA cathodes and phosphate and iron in LFP cathodes as each of them accounts for less than 2% of battery costs and their prices are much lower and less volatile compared to the studied critical materials. The price of phosphorus has certainly fluctuated in the past¹⁸. Nevertheless, the direct impacts of such price spikes on the transport sector only come to bear when the cost of phosphorus (in this case) relative to other minerals in LFP batteries is high. Therefore, we do not consider the influence of phosphorus price change on LFP adoption in this study.

We agree that the competition between agricultural demand and LFP demand for phosphorus will intensify in the future because about 85% of all the phosphorus mined is used in fertilizers, 10% for animal feed supplements, and the remainder for other products¹⁹. Phosphorus is listed by the European Commission as a “Critical Raw Material” with a high supply risk²⁰ because most countries are reliant on phosphorus imports to meet their agriculture and food demands. Reliable supply may be insufficient to meet the demand in the short- or long-term due to trade barriers, political insecurity and other supply chain disruption factors, which could cause price soaring risk of phosphorus in the future. Failure to consider the complexity of the global anthropogenic phosphorus cycle in the context of supply chain resilience and sustainability in the emerging EV industry is a potential factor influencing LFP adoption. Future research can incorporate LFP phosphorus projections into the global phosphorus cycle and trade context to ensure minimal potential conflict between future energy and food systems. Therefore, we added this constraint of LFP adoption in the Limitations part of the manuscript (lines 591-595).

In the main text:

Page 21, Lines 561-566:

Second, we do not consider the impact of phosphorus price changes on EV penetration when considering the adoption of LFP batteries given the negligible share of phosphorus in battery cost. However, the surging interest in LFP combined with the rising demand for phosphate from agriculture, the price of phosphorus (and other critical minerals) may move up along a non-stationary path and thus deserves further investigation in future research.

Supplementary Information Page 3 Line 55. I couldn't see it, but does the model take into account

the concomitant drop in the prices of fossil fuels. There may be a number of factors that affect this and the effects may be distributed differently in different geographies. The West's sanctions on Russia have affected fossil fuel prices, however China's lack of sanctions may mean China has access to cheap energy from Russia. Furthermore, as EVs displace ICEVs, presumably demand for petroleum products will begin to slow and in the absence of additional levies / taxes, there will presumably be a change in the demand - supply balance. This will further exacerbate the life-cycle cost differential between ICEVs and EVs.

Response:

We take prices changes of fossil fuels into consideration, which is endogenously calculated by GCAM.

Because our study focuses on the medium- and long-term changes of EV penetration rate in the road transport sector, factors that affect fuel costs in the short-term, such as West's sanctions on Russia or some other events, are not the part of our consideration.

Our results show that the cost differential between ICEVs and EVs will intensify as EVs gradually replace ICEVs to dominate the market. The life-cycle costs, which already includes fuel prices and non-fuel costs (vehicle costs), of EV and ICEV from 2020 to 2060 are shown in Figures S5-S11 and S28.

Despite the above reasoning, we agree that there is a need to add discussions on those factors which affect the life-cycle cost differential between ICEVs and EVs (such as competing with other low-carbon technologies for critical materials, the material needed for EVs being mined majorly as a byproduct of other materials, a limited number of suppliers, or the positioning of those suppliers in geopolitically unstable regions) in the revised manuscript (lines 357-376).

In the main text:

Pages 14-15, Line 351-370:

In addition to the input requirement of EV development, critical materials are also needed for other low-carbon technologies. Examples include neodymium, dysprosium, and praseodymium in wind power generation⁴¹; germanium, tellurium, indium, gallium, and manganese in solar power generation^{42,43}; nickel, cobalt, lithium, and platinum in fuel cell^{2,44}, and uranium, tungsten, tantalum, and molybdenum in nuclear energy⁴⁵. This means that the EVs sector has to compete with other low-carbon technologies for critical materials. It is highly likely that this competition will push up the prices of these critical materials far beyond our current expectations. What makes the competition tougher is that a number of these materials are concentrated in a few countries in politically volatile regions and produced by a handful number of companies⁴⁶⁻⁴⁸. Geopolitical tensions and socioeconomic unrests in the producing regions would disturb the material supply and result in significant price volatility^{49,50}. For example, cobalt is mined mainly as a by-product of nickel and copper, with approximately 71% of production and 51% of reserves concentrated in the Democratic Republic of Congo (DRC)⁵¹. In 2018, a policy shift in the country triggered an economic cascade that suspended the operations of Glencore's Mutanda mine, one of the DRC's largest cobalt mines. Whereafter the government announced to increase its mining royalty from 2% to 10%, price turbulence followed as a consequence³⁰. The ongoing Ukraine-Russia crisis has also brought additional volatilities to the supply of critical materials⁵². How to ensure the supply security of critical materials is a great challenge to the EV sector in China and beyond.

Page 11 Line 280. I see the section about the ongoing geopolitical tensions with Russia / Ukraine and the effect that this may have on Critical Raw Materials. Given the Chinese context, perhaps it is worth saying that given China's stance on the issue and lack of any sanctions, China may in effect be a net beneficiary of this situation, as companies in the West cancel trades with Russian metals firms, Nornickel e.t.c.

Response:

Thank you for your suggestions. Geopolitical tensions and socioeconomic unrests in the producing regions are important factors contributing to significant price volatility of critical materials. The ongoing Ukraine-Russia crisis (may be short-term, or other future events of conflicts) also demonstrates that the supply of critical materials is highly uncertain in the future. Thus, having a stable supply chain may be an important measure for China to ensure stable prices for critical materials. Accordingly, we revised this information in the manuscript line 373-376.

In the main text:

Page 14 Lines 368-370:

The ongoing Ukraine-Russia crisis has also brought additional volatilities to the supply of critical materials⁵². How to ensure the supply security of critical materials is a great challenge to the EV sector in China and beyond.

Page 12 308 – 313. It is perhaps worth explaining that recycled content from manufacturing scrap is available relatively quickly, however recycled material from end of life batteries is likely to take some time to return into the cycle, and so may not be available for some time. I suppose that there is also an implication here, that our patterns of consumption of private mobility do not change. It may be worth a comment, that given the constraints around Critical Material sourcing, other policy measures may need to be taken to increase the intensity with which we make use of extracted resources. Social fixes like product-service systems and the "uberisation" of vehicles, may allow us to serve more users using less vehicles in a resource constrained scenario, as private vehicles are a poorly optimised asset spending most of their time parked. Perhaps this study points to the need for unconventional solutions and public policy interventions as business as usual with ICEVs cannot continue.

Response:

We do agree that recycled material from end-of-life batteries is likely to take some time to return into the cycle, and thus may not be available within a short time span. The combination of closed-loop recycling and open-loop recycling may be a solution to the temporality concern of recycling. Open-loop recycling system, using other secondary sources such as industrial byproducts and wastes, can inherently reduce supply risk because there is reduced dependence on the primary suppliers and an increased number of suppliers overall. In addition, secondary sources are less geographically concentrated³⁴. But there are problems with open-loop recycling, for example, nickel recovered from stainless steel is not in suitable quality for batteries due to the high iron content. We have added an analysis of closed-loop and open-loop material recovery in the manuscript (lines 413-422).

Shared mobility schemes may help ease the growing desire for vehicle ownership and usage, thus indirectly reducing the demand for critical materials. We have added a paragraph to discuss the potential of the shared mobility schemes and the ways to promote the schemes (lines 456-467).

In the main text:

Page 15 Line 392-402:

Recycling is promising in addressing long-term critical material price challenges, as technological developments and economies of scale will reduce recycling costs. While recycling shortens supply chains and reduces logistical costs, at present it is still less expensive to mine the minerals than to recycle them, therefore, discovering processes for recovering valuable minerals which are cheaply enough to compete with newly mined minerals is urgently needed⁵⁹. Open-loop secondary sources may be an ideal choice (e.g., manufacturing scrap) to meet the challenges of closed-loop material recycling (e.g., the technical constraints), as secondary sources are often more widely distributed across geographical space⁶⁰. Nonetheless the grade of recycled material may require special attention. For example, nickel recovered from stainless steel is typically not in suitable quality for batteries due to the high iron content.

Page 16 Line 434-445:

Fourth, shared mobility schemes may help ease the growing desire for vehicle ownership and usage, thus indirectly reducing the demand for critical materials. Shared mobility schemes have the potential to reduce both personal vehicle usage and rates of ownership, which allows to serve more users using less vehicles in a resource constrained world⁶⁸. Recent research evidence shows that experience of using car-sharing has significant influence on decreasing the likelihood of choosing to use privately owned travel tools, such as private car^{69,70}. Therefore, government agencies and private-sector transport operators need to work together to develop attractive pricing models, combined with awareness campaigns to encourage consumers to better participate in and understand shared mobility schemes. The sequent snowball effect would help cities reap the huge potential benefits of these new forms of mobility and help the EV sector to better cope with the constraint of material scarcity.

Page 16 Line 413 On. I understand the limitations on many other materials. I'd perhaps question why Phosphate Rock isn't amongst the materials under evaluation given its prominence in LFP which is likely to become an increasingly dominant cathode chemistry.

Response:

In terms of cost, the cathode accounts for about 17% of the LFP cost, of which iron phosphate accounting for about 20% of cathode cost (with phosphorus accounting for about 53% of iron phosphate cost)¹⁷. The assumptions in this study neglect changes in the cost of any "other" cathode active materials including aluminum in NCA cathodes and phosphate and iron in LFP cathodes as each of them accounts for less than 2% of battery costs and their prices are much lower and less volatile than the four critical materials. Nevertheless, with the increasing adoption of LFP, it is also necessary to assess the impact of price changes in phosphate on the adoption of EVs in the future. Therefore, it is a limitation of this research that it does not consider phosphate price change (lines 561-566).

In the main text:

Page 21, Line 561-566:

Second, we do not consider the impact of phosphorus price changes on EV penetration when considering the adoption of LFP batteries given the negligible share of phosphorus in battery cost. However, the surging interest in LFP combined with the rising demand for phosphate from

agriculture, the price of phosphorus (and other critical minerals) may move up along a non-stationary path and thus deserves further investigation in future research.

My real query here is that my understanding is that changes in ICEV prices are modelled by on the flip side the total-cost of ownership of ICEVs is not. If consumers are making a choice between competing technologies, is it assumed that the prices of one ICEV stays relatively constant? I am not sure if e.g. fuel becomes cheaper if more pivot to EVs as there is less demand for Hydrocarbon fuels in transportation, or whether oil cartels will crimp output accordingly to maintain prices? Also... when we get to the point where EV vehicles are dominant, I wonder if the costs of maintaining the infrastructures for fossil fuels gets spread across a dwindling pool of consumers. I think for balance, the paper needs a section about how the total cost of ownership of ICEV vehicles will evolve in the transition.

Response:

The cost of ICEV is not constant over the forecast period. The total cost of ICEV is determined by fuel costs and non-fuel costs. The non-fuel cost (vehicle manufacturing cost) is exogenously determined according to the historical and forecasted development trend of ICEV cost in China as reported in the existing literature (summarized in Table S5). The fuel cost is endogenously determined by GCAM. The total cost of ICEV is shown in Fig. S12.

More demand for EVs means less demand for fossil fuels, which leads to cheaper fuels. As shown in Response letter Fig. R1 below, taking EVs equipped with NCM622 LIBs as an example, when the prices of critical minerals needed for EV rise, the penetration of EVs will decrease, leading to an increase in the demand for ICEVs, which will lead to a slight increase in fuel costs. By 2030, the fuel price of refined liquids will increase to 6.551 (1975)\$/GJ (High), 6.549 (1975)\$/GJ (Medium), and 6.546 (1975)\$/GJ (Low), which is 0.09%, 0.06%, and 0.01% higher than that in the BLS scenario, respectively. Thus, the price of refined liquids is higher in the material price surging scenarios than in the BLS scenario, meaning that more demand for EVs in the BLS leads to less demand for fossil fuels, thus cheaper fuels.

Fig. R1 Fuel price of refined liquids (1975)\$/GJ.

The infrastructure costs are a part of vehicle cost which is exogenously determined. Thus, when EVs are dominant, the infrastructures costs for fossil fuels will not get spread across a dwindling pool of consumers.

We have added the results on ICEV cost evolutions in Fig 1 and the first sub-section of the results (lines 153-164).

In the main text:

Page 5 Line 153-164:

It is the relative costs that influence the choice of consumers between competing technologies (e.g., ICEV). Therefore, we also analyze the evolution of ICEV cost (Figs. 1d and S13). Under BLS scenario, the ICEV costs in the LDV-4W sector are 0.042, 0.058, 0.098, and 0.153 (1990)\$/pass-km for a mini car, a subcompact car, a compact car, and a large car and SUV in 2020, and the increments of these values will be 4-12% by 2030 and 6-31% by 2060. The ICEV costs for light bus and heavy bus will be 0.027 and 0.043 (1990)\$/pass-km, respectively, by 2060, which are about 5% higher than those in 2030. The ICEV costs in the truck sector will increase by 1-11% between 2030 and 2060. The ICEV costs under the material price surge scenarios show slightly increases by about 0.02%-0.05% from the BLS, as a result of the increase in fuel costs caused by consumers switching from EVs to ICEVs. These results suggest that the ICEV cost will remain relatively stable during 2020 to 2060 under all scenarios.

a) Critical material price surge impact on EV development

b) Critical material price evolution

c) Electric vehicle (EV) cost evolution

d) Internal combustion engine vehicle (ICEV) cost evolution

Fig. 1. Evolution of critical material prices and costs of EVs and ICEVs from 2020 to 2060 under different scenarios. *Note:* **BLS** refers to the base-line scenario in which the uptake pace of EVs will fulfil the requirement of the carbon neutrality target and the EV cost will fall rapidly in line with its historical and forecasted development trend in China as reported in the existing literature; **High** scenario in which a rapid increase in critical material price affects EV costs; **Medium** scenario in which a steady increase in critical material price affects EV costs; **Low** scenario in which a slight increase in critical material price mainly affects EV costs during the middle and later periods of the forecast. All EVs in this figure are equipped with NCM622 LIBs; and the change in prices of critical materials is compared to the corresponding prices in 2015.

Reviewer #3:

Dear Laixiang Sun,

Thank you for the opportunity to review this paper. I found the analysis to be timely and helpful in framing the discussion of electric vehicle adoption in the face of potentially higher material costs.

My general comments and suggestions are as follows:

All the charts in Figure 4 are labeled as -high; I think these are supposed to be "high", "medium" and "low"?

Response:

Thank you for your suggestions, and sorry for the wrong labels. We have carefully checked the labels in the Figures and made corrections accordingly in Figures 4a, 4c, and 4e as High, Medium, and Low, respectively.

NMC111 is used as the example chemistry for discussion. I would suggest using NMC622 which is much more common for EVs now.

Response:

Thank you for your valuable suggestions. We now use NCM622 instead of NCM111 as the example chemistry for discussion. Accordingly, we have revised the related text (lines 125-139, 141-145, 183-189, 200-207, 214-221, 259-266, and Figs. 1, 2, and 3).

In the main text:

Pages 4-5, Lines 127-140:

Taking EVs equipped with NCM622 LIBs as an example, under the High scenario, the light duty vehicle-four wheels (LDV-4W) sector would have the highest increment in EV cost, which would reach 0.046, 0.070, 0.106, and 0.141 (1990)\$/pass-km for a mini car, a subcompact car, a compact car, and a large car and SUV by 2030 (7%, 9%, 10%, and 8% higher than those in the BLS scenario, respectively), with the corresponding cost figures reaching 0.048, 0.073, 0.116, and 0.148 (1990)\$/pass-km by 2060 (15%, 19%, 21%, and 18% higher than those in the BLS scenario, respectively). The EV cost in the bus sector will also increase sharply, making the cost of the light bus and heavy bus reach 0.023 and 0.040 (1990)\$/pass-km by 2060 (11% and 15% higher than those in the BLS). The electric trucks sector would have the lowest increment of about 9% compared to the BLS scenario by 2060. Meanwhile, the EV cost under the Medium scenario would be about 3-12% lower than that in High scenarios by 2060, due to the relatively lower level of threat by material price surge. The extent of EV cost increases will also be further reduced under the Low scenario, but from 2035 onwards, it would be 1-6% higher than those in the BLS.

Page 5 Line 143-147:

The costs of those EVs equipped with other types of LIBs will also be driven up by the price surges of critical materials, just like EVs with NCM622 LIBs (Figs. S5-S12). Under the High scenario, their costs would continue to rise, especially after 2035, with a relatively high increase for EVs equipped with ternary LIBs (5-14% and 5-32% higher than those in the BLS scenario by 2030 and 2060, respectively).

a) Critical material price surge impact on EV development

b) Critical material price evolution

c) Electric vehicle (EV) cost evolution

d) Internal combustion engine vehicle (ICEV) cost evolution

Fig. 1. Evolution of critical material prices and costs of EVs and ICEVs from 2020 to 2060 under different scenarios. *Note:* **BLS** refers to the base-line scenario in which the uptake pace of EVs will fulfil the requirement of the carbon neutrality target and the EV cost will fall rapidly in line with its historical and forecasted development trend in China as reported in the existing literature; **High** scenario in which a rapid increase in critical material price affects EV costs; **Medium** scenario in which a steady increase in critical material price affects EV costs; **Low** scenario in which a slight increase in critical material price mainly affects EV costs during the middle and later periods of the forecast. All EVs in this figure are equipped with NCM622 LIBs; and the change in prices of critical materials is compared to the corresponding prices in 2015.

Page 7 Line 186-191:

As shown in Fig. 2b, there would be around 37 million (High), 43 million (Medium), and 65 million (Low) units of EV in China by 2030, which are 44%, 35%, and 1% lower than those in the BLS in 2030, and these shares will decrease to 29%, 12%, and 2% by 2060. The increase

in the cost of EVs makes ICEVs more economically attractive. As a result, the total stock of ICEVs would reach 204 million (High), 197 million (Medium), and 181 million (Low) units by 2030, which is 6, 5, and 3 times the corresponding EV stock, respectively (Fig. 2c).

Fig. 2. Projections of vehicle stocks through 2020 to 2060 under different scenarios: a) total vehicle stocks, b) EV stocks by sub-sector, c) ICEV stocks by sub-sector. *Note:* The scenarios of **BLS**, **High**, **Medium**, and **Low** are the same as in Fig. 1; All EVs in this figure are equipped with NCM622 LIBs.

Page 8 Line 203-211:

Taking EVs equipped with NCM622 LIBs as an example, the EV penetration rate would decline to 35%, 41%, and 43% by 2030 under the High, Medium, and Low scenarios, respectively. Due to the continuous surge in the prices of critical materials, the resulting penetration rates of EVs under the High, Medium, and Low scenarios would be reduced to 51%, 60%, and 66%, respectively, in 2060, being 24%, 11%, and 1% lower than those under the BLS. With the increase in EV cost, the penetration rate of ICEVs will increase by 14 and 16 percentage points under the High scenario compared with the BLS by 2030 and 2060, respectively (Fig. S24).

Page 8 Line 217-225:

Our results show that the penetration rate of EVs would be 5, 12, and 13 percentage-points higher under the NCM622-High, NCM811-High, and NCM9.5.5-High scenarios than that in the NCM111-High scenario by 2030, and the corresponding values would increase to 22, 33, and 37 percentage-points by 2060. This means that replacing costly cobalt with nickel in LIBs can improve the market competitiveness of EVs. The cobalt-free LIBs will only increase the ICEV penetration rate by about 5 (2030) and 3 (2060) percentage-points under the LFP-High and LMO-High scenarios, and the resultant ICEV penetration rate are 9 and 13 percentage-points lower than those in the NCM622-High scenario, respectively.

Fig. 3. EVs and ICEVs penetration rate. *Note:* The scenarios of **BLS**, **High**, **Medium**, and **Low** are the same as in Fig. 1.

Page 10 Line 263-270:

Our result shows that when EVs are equipped with NCM622 LIBs, the cumulative CO₂ emissions in road transportation could reach 23, 21, and 19 Gt under the High, Medium, and Low scenarios, respectively, which will be 28%, 12%, and 5% higher than those under the BLS scenario. These values are 3.6, 3.2, and 3.0 times the carbon budget of the road transport sector, and 3.0, 2.6, and 2.5 times the transport carbon budget. EVs with cobalt-free LIBs (LFP and LMO) could reduce cumulative CO₂ emissions from road transport to a level of about 19 Gt, meaning a decrease by 3%-14% compared with the NCM622 scenario.

Prices for materials are likely to be linked (i.e., as low cobalt batteries are adopted, the price of cobalt may decrease and the price on nickel may correspondingly increase). This is difficult to capture in the methodology used in the paper, but should be discussed in the limitations section.

Response:

Thank you for the suggestion. We have added a brief discussion on the linked prices for materials in the limitation sub-section (lines 569-572).

In the main text:

Page 21 Line 569-572:

Third, it is worth noting that this study does not fully capture price linkages between materials, for example, cobalt prices may decrease with the adoption of low-cobalt batteries and nickel prices may increase accordingly, or lithium prices may increase further with the adoption cobalt-free LFP batteries.

The analysis also does not account for the effect of increasing availability and choice in electric vehicle models, which is likely to impact adoption independent of price. Another limitation of the study.

Response:

Thank you for the suggestion. We have added a brief discussion on this limitation (lines 566-569).

In the main text:

Page 21, Lines 566-569:

Although the price of critical materials is a significant factor affecting the penetration of EVs, we cannot ignore the influence of other factors on the adoption of EVs (e.g., increasing availability and choice in EV models), which should also be paid attention to in future research.

Although recycling is mentioned in the abstract and discussion, it is not discussed quantitatively in the paper. What is the percentage decrease in prices assumed to be attributable to recycling in the 3 scenarios?

Response:

To provide a dynamic account of the effects of material recycling on EV development, we couple a lifetime distribution delay forecasting model with the material flow analysis in the revision. The simulation analysis considers the second use and lifetime of batteries. This extension of the study confirms the importance of EV battery recycling in promoting sustainable transportation development under the condition of material scarcity constraint, suggesting that there will be about 10 years to witness the booming material recycling after 2030. Therefore, we have reason to believe that the recycling of LIBs will be timely in addressing the long-term material challenge. Accordingly, we add the relevant results (one subsection in Results) and discussions as follows:

In the main text:

Page 1 (Abstract), Lines 28-30:

Material recycling and technical innovation of lithium-ion batteries (LIBs) are effective countermeasures to cope with the material cost surging challenge, especially in the long-term.

Pages 12-13, Line 281-336:

Material Recycling Promotes Fleet Electrification

Fig. 5 report the results under the combinations of the RE (recycling) scenario and the High, Medium, and Low scenarios. The recycling potential of materials shows an increasing upward trend (Fig. 5a). Due to the delayed effects of material recycling, the resulting proportion of recycled materials to the total material demand in the LDV-4W, bus, and truck sector will be only 3%, 18%, and 3%, respectively, in 2030, however, this value could reach 85%, 86%, and 70%, respectively, by 2060. The recycled materials reduce the extent to which the materials needed for EVs are exposed to material price surges on international markets, thus reducing

the likelihood of cost surging for EVs. Taking EVs equipped with NCM622 LIBs as an example (Figs. 5a and S33), the EV cost will decrease to about 0.05-0.15 (1990)\$/pass-km by 2030 for LDV-4W under High-RE, Medium-RE, and Low-RE scenarios, which are slightly lower than those in High, Medium, and Low scenarios, respectively. In the bus and truck sector, material recycling will only help decrease the EV cost by about 1% by 2030. But the benefits of material recycling can be significant in the long-term. By 2060, the cost of EVs in the High-RE, Medium-RE, and Low-RE scenarios will be 11-15%, 4-6% and 4-5% lower than those in the High, Medium, and Low scenarios in the LDV-4W sector to reach about 0.05-0.13 (1990)\$/pass-km, which is basically the same as that in the BLS scenario (even in the High scenario). The EV cost in bus and truck sector will have about 2-10% decrease in RE scenarios by 2060. These results clearly manifest that material recycling can greatly reduce the impact of surging material prices on the EV cost, especially in the long-term.

The decrease in EV cost will raise EV penetration rate. As shown in Fig. 5b, the material recycling will help increase the EV (with NCM622 LIBs) penetration rate by 7, 1, and 1 percentage points by 2030 in High-RE, Medium-RE, and Low-RE scenarios, respectively, and the resultant rates are still 14%, 8%, and 6% lower than those in BLS scenario. However, the recycling will boost the EV uptake rate to 59%, 66%, and 67% in High-RE, Medium-RE, and Low-RE scenarios, respectively, by 2060 (much closer to 67% under the BLS). This will inevitably reduce ICEV's market share by 12 percentage points (to about 10%), 2 percentage points (to about 6%), and 2 percentage points (to about 6%), making ICEV's market share close to that in the BLS (6%). Let all EVs be equipped with cobalt-free LIBs, recycling can reduce ICEV market penetration to baseline levels even in the High-RE scenario (other RE-combining scenarios than NCM622-RE can be found in Figs. S39-45). These results highlight that recycling would have remarkable effects to mitigate the material price challenge in EV development in the long-term.

Due to the positive role of material recycling in promoting EV development, the resulting cumulative CO₂ emissions from road transportation in 2020 to 2060 under High-RE, Medium-RE, and Low-RE scenarios decrease to 22 Gt, 20 Gt, and 19 Gt, respectively, which are 8%, 2%, and 1% lower than those under High, Medium, and Low scenarios, respectively (17%, 9%, and 4% higher than those in BLS scenario) (Fig. 5c). Although the CO₂ emissions from road transportation in the RE-combining scenarios will only decrease by less than 1% compared with that in High, Medium, and Low scenarios by 2030. After 2030, the differences in CO₂ emissions between the RE and BLS scenarios become narrowing. By 2060, the CO₂ emissions will decrease to 0.27 Gt/yr, 0.24 Gt/yr, and 0.23 Gt/yr in High-RE, Medium-RE, and Low-RE scenarios, respectively, which are 36%, 12%, and 10% lower than those in High, Medium, and Low scenarios (Fig. 5d). For EVs with cobalt-free LIBs, materials recycling can reduce the cumulative CO₂ emissions to a level only 2-7% higher than those under the BLS (Fig. S46). This indicates that materials recycling can facilitate low-carbon transition in the transportation sector in the long-term.

Fig. 5. Material recycling effects on EV development and CO₂ emissions of road transportation in China from 2020 to 2060 under the RE scenario: a) Material recycling potential and the effect on EV cost evolution; b) EV and ICEV penetration rate, c) cumulative CO₂ emissions, d) CO₂ emissions by year. *Note:* The scenarios of **BLS**, **High**, **Medium**, and **Low** are the same as in Fig. 1; RE is a scenario in which only primary demand is affected by the market price of the material concerned; MS is the market share of vehicles; All EVs in this figure are equipped with NCM622 LIBs.

Page 19, Lines 493-502:

Material flow analysis

Since GCAM does not count the number of vehicles explicitly, a conversion of transportation service demand into the number of vehicles is required. Eq (13) in Note S5 presents the conversion formula. We then adopt a stock-driven dynamic material-flow-analysis (MFA) model to estimate the inflow (sale) and outflow (decommissioning) of vehicles. The technical details are presented in Note S5.

Considering that battery's operational lifetime has a significant impact on material recycling and the EV adoption⁷⁴. We couple a lifetime distribution delay forecasting model with the dynamic MFA to investigate the effects of recycling on EV development, which considers the second use and lifetime of batteries. Please see Note S6 for technical details.

Page 20, Lines 544-550:

Recycling scenarios

In the recycling scenario (RE), we assume that the recycling is closed-looped, namely, the recycled minerals reaches the quality for battery production³⁷. The materials obtained by battery manufacturers through recycling are not affected by material price fluctuations on the international market, that is, only the primary demand for materials is affected by the surging material prices. The material recycling potential is calculated according to the material demand under the BLS scenario.

The analysis is likely to be sensitive to starting assumptions (e.g., the nominal value in the medium price scenario). The supplementary information should include a more in-depth discussion of how these values were derived and their impact on the final results.

Response:

Following this important suggestion, we have added the detailed descriptions regarding these starting assumptions in Note S2. The historical price dynamics of these critical materials, which underpinning the forecasting, are shown in Fig. S3. We have also carried out a sensitivity analysis by varying the price of critical minerals used in this study, and the results are shown in Fig. S30.

In the SI:

Note S2. Critical material price forecast

There is great uncertainty in the long-term prediction of material prices. Therefore, we use three methods to do the forecasts. The historical price dynamics of these critical materials, which underpinning the forecasting, are shown in Fig. S3.

In the High scenarios, we postulate that the initial surge of demand for EV will accelerate the rising of the prices of critical materials but the acceleration will dampen in the medium-run and the price level will become flatten in the long-run thanks to the increased recycling and the increased use of substitutes. We use a logistic function to predict the prices of critical materials, in which the relationship between price and demand quantity is an S-shaped curve lying between the lower and upper limit of the price (Eq. 9)³:

$$\frac{P-P_L}{P_U-P_L} = \frac{1}{1+\exp(-c_1-c_2D)} \quad (9)$$

where P_L is the lower limit of material price, which is derived from historical data; P_U is the upper limit of material price, which we estimate to be multiple of the highest price observed in the history; D is the quantity of annual material demand; and c_1 and c_2 are coefficients, which are estimated by regressions based on historical demand data.

In the Medium scenarios, we use consumer price index (CPI) to deflate and then use regression analysis to forecast the future dynamics of material prices⁴. It is common knowledge that the purchasing power of a dollar in 1850 is significantly higher than that of a dollar today. The extent of changes in the purchasing power of a dollar between a given year and the base year is measured by price indices. The CPI is the most commonly used price index to quantify the purchasing power of a dollar in a given year relative to the given base-year, which is based on the values of a basket of items a representative consumer would buy (like foods, housing, transport entertainment etc.) in the given year and the base-year.

Understanding how the price of the metal in question increases or decreases in relation to the price of a standard basket of goods will give better insight than looking at the nominal price in isolation. Therefore, we will use the CPI, which is released by the United States Department of Labor⁵, to remove the effect of inflation as presented in Eq. 10 below.

$$\text{RealValue}_{\text{year}_j} = \text{NominalValue}_{\text{year}_q} \cdot \frac{\text{CPI}_{\text{year}_j}}{\text{CPI}_{\text{year}_q}} \tag{10}$$

In the Low scenarios, we predict the long-term changes in metal prices based on the regression of logged prices on logged demand quantity (Single-Factor Learning Curve)⁵.

Figure S3. Historical and forecasted prices of critical materials

Figure S30. Sensitivity analysis for EV deployment in China in 2030. *Note:* EVs are equipped with NCM622 LIBs.

References

1. Kyle, P. & Kim, S. H. Long-term implications of alternative light-duty vehicle technologies for global greenhouse gas emissions and primary energy demands. *Energy Policy* **39**, 3012-3024 (2011).
2. Edelenbosch, O. Y. *et al.* Decomposing passenger transport futures: comparing results of global integrated assessment models. *Transport. Res. Part D-Transport. Environ.* **55**, 281-293 (2017).
3. Fulton, L., Cazzola, P. & Cuenot, F. IEA Mobility Model (MoMo) and its use in the ETP 2008. *Energy Policy* **37**, 3758-3768 (2009).
4. Façanha, C., Blumberg, K. & Miller, J. Global transportation energy and climate roadmap. *International Council on Clean Transportation* (2012).
5. Hao, H., Wang, H. & Yi, R. Hybrid modeling of China's vehicle ownership and projection through 2050. *Energy* **36**, 1351-1361 (2011).
6. McJeon, H. *et al.* Limited impact on decadal-scale climate change from increased use of natural gas. *Nature* **514**, 482-485 (2014).
7. Shukla, P. R. & Chaturvedi, V. Low carbon and clean energy scenarios for India: Analysis of targets approach. *Energy Econ.* **34**, S487-S495 (2012).
8. Milovanoff, A., Posen, I. D. & MacLean, H. L. Electrification of light-duty vehicle fleet alone will not meet

- mitigation targets. *Nat. Clim. Change* **10**, 1102-1107 (2020).
9. McCollum, D. L. *et al.* Interaction of consumer preferences and climate policies in the global transition to low-carbon vehicles. *Nat. Energy* **3**, 664-673 (2018).
 10. Isik, M., Dodder, R. & Kaplan, P. O. Transportation emissions scenarios for New York City under different carbon intensities of electricity and electric vehicle adoption rates. *Nat. Energy* **6**, 92-104 (2021).
 11. Baars, J., Domenech, T., Bleischwitz, R., Melin, H. E. & Heidrich, O. Circular economy strategies for electric vehicle batteries reduce reliance on raw materials. *Nat. Sustainability* **4**, 71-79 (2021).
 12. Hao, H. *et al.* Impact of transport electrification on critical metal sustainability with a focus on the heavy-duty segment. *Nat. Commun.* **10**, 1-7 (2019).
 13. Peng, T., Ou, X., Yuan, Z., Yan, X. & Zhang, X. Development and application of China provincial road transport energy demand and GHG emissions analysis model. *Appl. Energy* **222**, 313-328 (2018).
 14. Pan, X., Wang, H., Wang, L. & Chen, W. Decarbonization of China's transportation sector: in light of national mitigation toward the Paris Agreement goals. *Energy* **155**, 853-864 (2018).
 15. Khanna, N., Lu, H., Fridley, D. & Zhou, N. Near and long-term perspectives on strategies to decarbonize China's heavy-duty trucks through 2050. *Sci Rep* **11**, 1-14 (2021).
 16. Abdelbaky, M., Peeters, J. R. & Dewulf, W. On the influence of second use, future battery technologies, and battery lifetime on the maximum recycled content of future electric vehicle batteries in Europe. *Waste Manage.* **125**, 1-9 (2021).
 17. *Report on In-depth Research and Investment Prospect Forecast of China Lithium Iron Phosphate Cathode Material Industry (2022-2029) (in Chinese)* (INSIGHT AND INFO, 2022).
 18. Baffes, J. & Koh, W. Soaring Fertilizer Prices Add to Inflationary Pressures and Food Security Concerns. *World Bank, Data Blog* (2021).
 19. Spears, B. M., Brownlie, W. J., Cordell, D., Hermann, L. & Mogollón, J. M. Concerns about global phosphorus demand for lithium-iron-phosphate batteries in the light electric vehicle sector. *Commun. Mater.* **3**, 1-2 (2022).
 20. *Critical Raw Materials Resilience: Charting a Path towards Greater Security and Sustainability* (European Commission Brussels, Belgium, 2020).
 21. Habib, K. & Wenzel, H. Exploring rare earths supply constraints for the emerging clean energy technologies and the role of recycling. *J. Cleaner Prod.* **84**, 348-359 (2014).
 22. Nassar, N. T., Wilburn, D. R. & Goonan, T. G. Byproduct metal requirements for US wind and solar photovoltaic electricity generation up to the year 2040 under various Clean Power Plan scenarios. *Appl. Energy* **183**, 1209-1226 (2016).
 23. Zuser, A. & Rechberger, H. Considerations of resource availability in technology development strategies: The case study of photovoltaics. *Resour. Conserv. Recycl.* **56**, 56-65 (2011).
 24. Hao, H. *et al.* Securing platinum-group metals for transport low-carbon transition. *One Earth* **1**, 117-125 (2019).
 25. Giurco, D., McLellan, B., Franks, D. M., Nansai, K. & Prior, T. Responsible mineral and energy futures: views at the nexus. *J. Cleaner Prod.* **84**, 322-338 (2014).
 26. Chitre, A., Freake, D., Lander, L., Edge, J. & Titirici, M. M. Towards a More Sustainable Lithium-Ion Battery Future: Recycling LIBs from Electric Vehicles. *Batteries Supercaps* **3**, 1125-1125 (2020).
 27. Erdmann, L. & Graedel, T. E. Criticality of non-fuel minerals: a review of major approaches and analyses. *Environ. Sci. Technol.* **45**, 7620-7630 (2011).
 28. Chu, S. & Majumdar, A. Opportunities and challenges for a sustainable energy future. *Nature* **488**, 294-303 (2012).

29. Craighead, C. W., Blackhurst, J., Rungtusanatham, M. J. & Handfield, R. B. The severity of supply chain disruptions: design characteristics and mitigation capabilities. *Decis. Sci.* **38**, 131-156 (2007).
30. Alonso, E., Gregory, J., Field, F. & Kirchain, R. *Material availability and the supply chain: risks, effects, and responses* (ACS Publications, 2007).
31. Gulley, A. L., McCullough, E. A. & Shedd, K. B. China's domestic and foreign influence in the global cobalt supply chain. *Resour. Policy* **62**, 317-323 (2019).
32. Akcil, A., Sun, Z. & Panda, S. COVID-19 disruptions to tech-metals supply are a wake-up call. **587**, 365-367 (2020).
33. Kinch, D. Metals prices may rally further if Russia-Ukraine tensions impact supply: analysts. *S&P Global Commodity Insights* <https://www.spglobal.com/commodity-insights/en/market-insights/latest-news/energy-transition/012122-metals-prices-may-rally-further-if-russia-ukraine-tensions-impact-supply-analysts> (2022).
34. Bustamante, M. L. & Gaustad, G. Challenges in assessment of clean energy supply-chains based on byproduct minerals: A case study of tellurium use in thin film photovoltaics. *Appl. Energy* **123**, 397-414 (2014).
35. Gaines, L. The future of automotive lithium-ion battery recycling: Charting a sustainable course. *Sustainable Mater.Technol.* **1**, 2-7 (2014).
36. Zhou, F. *et al.* Examining the impact of car-sharing on private vehicle ownership. *Transp. Res. Pt. A-Policy Pract.* **138**, 322-341 (2020).
37. Zhou, F. *et al.* Preference heterogeneity in mode choice for car-sharing and shared automated vehicles. *Transp. Res. Pt. A-Policy Pract.* **132**, 633-650 (2020).

REVIEWERS' COMMENTS

Reviewer #3 (Remarks to the Author):

Thank you for the opportunity to review this paper. The work provides a valuable economic perspective on the value of recycling EV batteries. The authors addressed my previous comments thoroughly and thoughtfully. I recommend publishing this paper.